# Trustworthy Machine Learning through Data-Specific Indistinguishability

**Hanshen Xiao** [1 2]  **Zhen Yang** [2]  **G. Edward Suh** [1]

## Abstract

This paper studies a range of AI/ML trust concepts, including memorization, data poisoning, and copyright, which can be modeled as constraints on the *influence* of data on a (trained) model, characterized by the *outcome difference* from a processing function (training algorithm). In this realm, we show that provable trust guarantees can be efficiently provided through a new framework termed *Data-Specific Indistinguishability* (DSI) to select trust-preserving randomization tightly aligning with targeted outcome differences, as a relaxation of the classic *Input-Independent Indistinguishability* (III). We establish both the theoretical and algorithmic foundations of DSI with the optimal multivariate Gaussian mechanism. We further show its applications to develop trustworthy deep learning with *black-box* optimizers. The experimental results on memorization mitigation, backdoor defense, and copyright protection show both the efficiency and effectiveness of the DSI noise mechanism.

## 1. Introduction

The remarkable advances in artificial intelligence (AI) and machine learning (ML) in recent years have been largely driven by optimizing loss functions based on predictive accuracy. Despite this success, the widespread deployment of AI has raised a range of societal and ethical concerns, particularly related to the use of data and its impact on trained models. These concerns have spurred the emergence of research on trustworthy data processing.

In this paper, we present a unified and systematic framework for a critical subset of trust concepts. These concepts can be modeled as constraints on the differences between the outputs of an algorithm (a machine learning training algorithm) for a given input (dataset) and an additional set of

reference inputs (datasets). We formalize the framework of Differential Trust (DT) as follows. Given a processing function $\mathcal{F}$, a target input $\mathsf{U}$, and a group of reference datasets $\mathsf{R}_{[1:m]} = \{\mathsf{R}_1, \cdots, \mathsf{R}_m\}$, (some of) the outputs $\mathcal{F}(\mathsf{R}_i)$ derived from the reference datasets are assumed to be safe; our objective is to regulate the divergence between the output $\mathcal{F}(\mathsf{U})$ generated from $\mathsf{U}$ and the outputs $\mathcal{F}(\mathsf{R}_i)$ from the references. Several concrete examples are listed below.

**Memorization**: Unintended memorization is a common issue in large models, where the model learns specific details or features from individual examples rather than capturing the common patterns of the underlying data distribution (Tirumala et al., 2022). In the context of language models (LMs), mitigating memorization typically involves defending against queries that attempt to regurgitate sensitive training data verbatim (Zanella-Béguelin et al., 2020). For instance, (Tirumala et al., 2022; Carlini et al., 2022) define memorization behaviors as follows: given a concatenated context from the input training set in a form $[p|s]$, the LM is able to exactly reproduce $s$ when prompted with $p$. Suppose $[p|s]$ is uniquely contained in a sample $u \in \mathsf{U}$, and $\mathcal{F}$ is a training algorithm. By selecting a reference $\mathsf{R} = \mathsf{U} \setminus u$, the resulting model $\mathcal{F}(\mathsf{R})$ can be considered safe against the memorization of $[p|s]$, as this feature is never seen by the model during training (Carlini et al., 2021).

**Data Poisoning (Backdoor) Attack**: By injecting specially-crafted training data, an attacker can manipulate the training process to embed hidden, malicious behavior into the model while maintaining its performance on normal data. For example, adding a specific patch to an image might cause the model to misclassify it when a particular sticker is present (Gu et al., 2017; Tran et al., 2018). Existing empirical defense largely relies on filtering suspicious samples (Jagielski et al., 2018; Baracaldo et al., 2017) or robust gradient aggregator (Tran et al., 2018; Kane et al., 2024), while providing provable guarantees remains challenging. In the DT framework, consider a (virtual) distributed setup where $\mathsf{U}$ consists of data collected from $m$ sources, with one of them being poisoned. Let $\mathsf{R}_i$ represent the subset of $\mathsf{U}$ after excluding all data provided by the $i$-th source. Among the $m$ models $\mathcal{F}(\mathsf{R}_i)$, $i \in [1 : m] = \{1, 2, \ldots, m\}$, there is guaranteed to be at least one model trained entirely on benign data.

**Copyright and Contribution**: Ensuring training samples

[1]NVIDIA [2]Department of Computer Science, Purdue University. Correspondence to: Hanshen Xiao <hsxiao@purdue.edu>.

*Proceedings of the 42nd International Conference on Machine Learning*, Vancouver, Canada. PMLR 267, 2025. Copyright 2025 by the author(s).

on an Internet scale entirely free of copyrighted materials is often impractical (Vyas et al., 2023). In addition, the recent success of generative models has raised growing concerns, particularly among the artist community where generated images or music can closely mimic specific artistic styles, potentially infringing on intellectual property rights (Shan et al., 2023; Dhariwal et al., 2020). Furthermore, in collaborative learning (Sim et al., 2020) where a model is trained using data from multiple parties, how to quantify their individual contributions to acknowledge ownership or determine compensation remains open. In these use cases, a reference R can be selected as the subset of U that excludes all samples from a particular party (e.g., an artist, author, or data source). Consequently, $\mathcal{F}(R)$ serves as a reference that excludes influence from the particular party's data.

## 1.1. Indistinguishability

To build provable DT guarantees, we aim to ensure our target output $\mathcal{F}(U)$ is *indistinguishable* from the safe references $\mathcal{F}(R_i)$, i.e., probabilistically, $\mathcal{F}(U)$ is distributed sufficiently close to *each* $\mathcal{F}(R_i)$. In the context of above examples, this presents a guarantee to either *simultaneously* protect copyright or prevent memorization for each differential data subset $U \setminus R_i$, or maintain robustness against backdoor attacks provided the model remains close to the one that is only trained with a benign dataset $R_i$.

Historically, the idea of (input-independent) indistinguishability traces back to Shannon's pioneering work on *perfecrt secrecy* (Shannon, 1949), which laid the foundation for both Differential Privacy (DP) (Dwork et al., 2006b) and modern cryptography (Goldwasser & Micali, 1984). We summarize this framework below.

**Definition 1.1** (($\bar{\gamma}, \psi$) Input-Independent Indistinguishability (III))**.** Given a divergence measure $\psi$, a mechanism $\mathcal{M} : \mathcal{U}^* \rightarrow \mathcal{O}^*$ satisfies ($\bar{\gamma}, \psi$)-III if for *arbitrary* two input selections $\bar{U}$ and $\bar{U}'$ differing in some objective feature,

$$\sup_{\bar{U}, \bar{U}'} \psi\big(\mathbb{P}_{\mathcal{M}(\bar{U})} \| \mathbb{P}_{\mathcal{M}(\bar{U}')}\big) \leq \bar{\gamma}. \tag{1}$$

(1) characterizes, given the output $\mathcal{M}(U)$ for some input U, the hardness to distinguish U from two *arbitrary* candidates $\bar{U}$ and $\bar{U}'$ in the *worst case*. By properly selecting the objective feature as the sensitive part of a secret input U to protect, Definition 1.1 can capture many classic privacy definitions. For example, DP aims to obscure the participation of an individual record and we may select $\bar{U}$ and $\bar{U}'$ as a pair of *adjacent datasets* differing in one datapoint; accordingly, when we select $\psi$ varying from maximal, Hockey-Stick and Rényi divergence, (1) leads to pure $\epsilon$-DP (Dwork et al., 2006b), approximate $(\epsilon, \delta)$-DP (Dwork et al., 2006a) and Rényi DP (Mironov, 2017). Recent works have also shown applications of DP to mitigate memorization (Carlini et al., 2021) or approximate unlearning (Gupta et al., 2021).

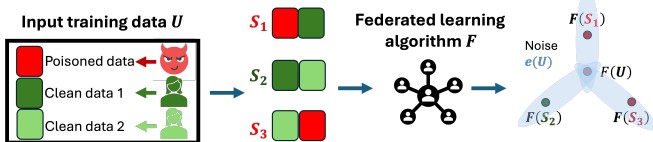

*Figure 1.* Illustration of applications of DSI noise in defending against poisoning attacks. Provided an input training data U collected from three data sources including an unknown malicious entity, one reference subset $S_2$ is fully clean; the injected noise $e(U)$ ensures that produced model $\mathcal{F}(U) + e(U)$ is statistically close to the noisy model $\mathcal{F}(S_2) + e(U)$ trained on fully clean data.

## 1.2. From Input-Independent Indistinguishability (III) to Data-Specific Indistinguishability (DSI)

Although both privacy protection — focused on confidentiality protection as in DP and PAC Privacy (Xiao & Devadas, 2023; Xiao et al., 2024; Sridhar et al., 2025) — and DT leverage indistinguishability, their underlying motivations are different. A key insight we will highlight in this work is that input independence is *not* necessary for many DT applications, where we only need indistinguishability with respect to a specific set of safe reference inputs, enabling much sharpened utility-trust tradeoff.

**When Input Independence is Needed**: A widely accepted approach to measuring privacy risk is to quantify the additional knowledge gained through leakage—specifically, how much the leakage alters the adversary's prior belief and aids their inference about sensitive information. III (Definition 1.1) requires satisfactory indistinguishability (1) holds for *arbitrary* input pairs. This is necessary if we want to upper bound the additional knowledge[1] for an adversary with *arbitrary* belief and optimal strategy (Shannon, 1949).

For confidentiality protection, to ensure a processing function $\mathcal{F}$ with satisfactory III guarantees, the introduced modification, (e.g., noise), should *not* leak information about the secret input U itself. Therefore, most III randomization approaches, including exponential mechanism (McSherry & Talwar, 2007) and Laplace/Gaussian noise mechanism (Dwork et al., 2014) in DP, are input-independent[2] and only determined by the public information of $\mathcal{F}$ to privatize.

As a global guarantee, clearly III shows a *sufficient* condition to ensure formal mitigation for DT. However, from a practice standpoint, it faces two challenges:

i) a worst-case bound (1) may fail to account for varying indistinguishability demands among references. For instance,

---

[1]Alternatively, the additional knowledge can be defined by the optimal posterior success rate of an adversarial inference on the sensitive information after observing the leakage minus the optimal *a priori* success rate (Xiao & Devadas, 2023).

[2]As a clarification, input-dependent randomness could still lead to III, for example, noise calibrated with smooth sensitivity (Nissim et al., 2007), but our proposed input-dependent Gaussian as defined later in Definition 1.2 generally cannot ensure III.

in the example of data contribution mentioned earlier, a user may wish to allocate different budgets based on data quality or preferences for the $i$-th provider's data usage;

ii) III is also notorious for its heavy utility trade-offs. The required level of randomness to achieve III has been extensively studied and is often constrained by the curse of dimensionality (Hardt & Talwar, 2010; Bassily et al., 2014; Xiao et al., 2023a), where the utility compromise increases with the dimensionality of the output.

**DSI and Noise Mechanism**: With above understanding, in applications where our inputs are public, for example artists are willing to publish and allow others to have access to their works once their intellectual property is properly protected, or in the scenarios where we do not target information leakage but the governance of AI models against specific adversarial strategies, for example memorization mitigation resisting prompt elicitation to reproduce specific input contexts, both $\mathcal{F}(\mathsf{U})$ and safe references $\mathcal{F}(\mathsf{R}_i)$ can be freely used as public parameters in mechanism designs, different from privacy-preserving operations. Additionally, we only require *Data-Specific Indistinguishability* (DSI) between each pair $(\mathcal{F}(\mathsf{U}), \mathcal{F}(\mathsf{R}_i))$ for $i = 1, 2, \cdots, m$.

Operationally, since a given (possibly black-box) processing function $\mathcal{F}$ may not inherently satisfy *all* differential requirements, we consider introducing a (multivariate) Gaussian noise $e$ for required DSI such that for each pair $(\mathsf{U}, \mathsf{R}_i)$, $i \in [1 : m]$, the noisy distributions $\mathcal{F}(\mathsf{U}) + e$ and $\mathcal{F}(\mathsf{R}_i) + e$ are sufficiently close, as formalized below.

**Definition 1.2** (($\gamma_{[1:m]}, \psi$) Data-Specific Indistinguishability (DSI) Gaussian Mechanism)**.** Given a processing function $\mathcal{F}$ and an input $\mathsf{U}$ with $m$ associated reference datasets $\mathsf{R}_{[1:m]}(\mathsf{U}) = \{\mathsf{R}(\mathsf{U})_i, i \in [1 : m]\}$ possibly determined by $\mathsf{U}$, for some divergence measure $\psi$, an $(\gamma_{[1:m]}, \psi)$ DSI Gaussian mechanism returns a zero-mean Gaussian distribution $\mathcal{N}(0, \Sigma(\mathsf{U}))$ whose covariance $\Sigma(\mathsf{U})$ can be selected dependent on input $\mathsf{U}$ and references $\mathsf{R}_{[1:m]}(\mathsf{U})$ such that for a noise $e(\mathsf{U})$ independently sampled from $\mathcal{N}(0, \Sigma(\mathsf{U}))$,

$$\psi(\mathbb{P}_{\mathcal{F}(\mathsf{U})+e(\mathsf{U})} \| \mathbb{P}_{\mathcal{F}(\mathsf{R}_i)+e(\mathsf{U})}) \leq \gamma_i, i = 1, \cdots, m, \quad (2)$$

where $\mathbb{P}_a$ represents the distribution of a random variable $a$.

The *data-dependent* or *data-specific* aspect in Definition 1.2 is two-fold. First, the DSI in (2) only requires satisfactory indistinguishability regarding specific output pairs produced by our target input $\mathsf{U}$ and references $\mathsf{R}_{[1:m]}(\mathsf{U})$. Second, both the references $\mathsf{R}_{[1:m]}(\mathsf{U})$ and the injected noise $e(\mathsf{U})$ can be adaptively selected dependent on the input $\mathsf{U}$. In (2), a smaller $\gamma_i$ implies a stronger mitigation on the differential effect between $\mathcal{F}(\mathsf{U})$ and $\mathcal{F}(\mathsf{R}_i)$ (their noisy versions are statistically closer and harder to distinguish), which typically requires a larger noise variance along the direction of difference $\mathcal{F}(\mathsf{U}) - \mathcal{F}(\mathsf{R}_i)$. An application of backdoor defense with DSI noise mechanism is illustrated in Fig. 1.

As a reasonable relaxation, DSI also systematically addresses the two above-mentioned challenges of applying III to DT: i) by taking the target input $\mathsf{U}$ and its references as parameters in the noise mechanism, we allow a user to freely adjust the distinguishability levels $\gamma_i$ across references; ii) from utility standpoint, the optimal DSI noise is *independent of dimensionality* and can be upper bounded by the number $m$ of references, which makes high-dimensional trustworthy processing feasible, as elaborated in Section 3.

### 1.3. Contribution and Paper Organization

In this paper, we initiate the study on Differential Trust (DT) through DSI-based defense with fourfold contributions from concept, algorithm, theory and application perspectives.

1. **DT and DSI Framework**: We formalize DT as a unified way to define a wide range of trust concepts and propose the concept of DSI to support quantitative studies and guarantees.

2. **Optimal Gaussian Mechanism**: We show, for $\psi$ being an *arbitrary* (non-decreasing function of) $f$-divergence, determining the high-dimensional Gaussian noise with minimal variance satisfying Definition 1.2 can be reduced to a convex optimization (Lemma 3.1) over $m$ non-negative Lagrangian multipliers (Theorem 3.3), which is efficiently solvable (Algorithm 1). Of independent interest, the proposed noise mechanism also addresses the operational challenge in controlling per-instance or individual privacy (Wang, 2019; Feldman & Zrnic, 2021; Thudi et al., 2024) (Appendix A).

3. **Properties of DSI**: We comprehensively study the properties of DSI guarantees, including post-processing immunity, probabilistic interpretation, composition (Theorem 3.4) and grouping (Lemma 3.5), to facilitate its application.

4. **DSI Deep Learning**: Stemmed from Differentially Private Stochastic Gradient Descent (DP-SGD) (Abadi et al., 2016), we propose a more general framework (Algorithm 2) to iteratively incorporate optimal DSI noise into a deep learning procedure with *black-box* optimizers. We show dimensionality-independent DSI noise and the algorithmic flexibility, which allows local updates *without* artificial clipping, significantly sharpen the utility compromise (Table 1) especially when scaling models. We show its applications in cutting-edge applications for various trust concepts, including memorization mitigation in large language models (LLM), resistance to backdoor attacks, and copyright protection (Section 5).

## 2. Preliminaries

**Divergence and Probability**: One possible selection of the measurement $\psi$ in Definition 1.2 is $f$-divergence, as defined below.

**Definition 2.1** ($f$-Divergence). Let $f : (0, +\infty) \to \mathbb{R}$ be a convex function with $f(1) = 0$. Let P and Q be two distributions on some measurable space, and the $f$-divergence $\mathcal{D}_f$ between P and Q is defined as

$$\mathcal{D}_f(\mathsf{P} \| \mathsf{Q}) := \mathbb{E}_Q\big[f(d\mathsf{P}/d\mathsf{Q})\big]. \tag{3}$$

Here, we use $d\mathsf{P}$ ($d\mathsf{Q}$) to represent the probability density (or mass) function of probability distribution P (Q).

Many commonly-used statistical divergence measurements are special cases of $f$-divergence. In Definition 2.1, when we select $f(t) = t\log(t)$, (3) becomes the Kullback-Leibler (KL) divergence; when $f(t) = |t - 1|/2$, (3) becomes the Total Variation (TV) $\mathcal{D}_{TV}(\mathsf{P} \| \mathsf{Q}) = 1/2 \cdot \int |d\mathsf{P} - d\mathsf{Q}|$.

$f$-divergence enjoys multiple desirable properties to facilitate both the interpretation and usage of DSI guarantees. As detailed in Appendix B, DSI guarantees measured in $f$-divergence is robust to *post-processing* (Lemma B.1). That is to say, suppose we have found a proper noise $e(\mathsf{U})$ such that (2) holds for the noisy version of $\mathcal{F}(\cdot)$, then the same security parameters still hold for the output $\mathcal{G}(\mathcal{F}(\cdot) + e(\cdot))$ after applying a post-processing operator $\mathcal{G}$. Other useful properties including *shift invarfiance* (Definition B.2) and *joint convexity* (Definition B.3), which will be used in determining the optimal noise. Additionally, many other divergences enjoying above-mentioned properties are essentially a non-decreasing function of some $f$-divergence. One representative is Rényi Divergence defined below.

**Definition 2.2** ($\alpha$ Rényi Divergence). For $\alpha > 1$, the $\alpha$ Rényi Divergence $\mathcal{R}_\alpha$ between two distributions P and Q with identical support sets is defined as

$$\mathcal{R}_\alpha(\mathsf{P} \| \mathsf{Q}) := 1/\alpha \cdot \log\big(\mathbb{E}_Q\big[(d\mathsf{P}/d\mathsf{Q})^\alpha\big]\big).$$

$\mathcal{R}_\alpha$ can be viewed as a logarithm of an $f$-divergence with selecting $f(t) = t^\alpha$. Divergences are also useful to bound the probability difference of some event in two distributions, and we take Rényi Divergence as an example.

**Lemma 2.3** (Probability Difference Bound (Langlois et al., 2014)). *For two distributions* P *and* Q *with identical support sets and an arbitrary event* A, *the probability*

$$\mathsf{P}(A) \le \mathsf{Q}(A)^{\frac{\alpha-1}{\alpha}} \cdot \big(\mathcal{R}_\alpha(\mathsf{P} \| \mathsf{Q})\big)^{1/\alpha}. \tag{4}$$

**Additional Notations**: For a positive semi-definite (PSD) matrix $C$, suppose its singular value decomposition (SVD) is in a form $C = P \cdot Q \cdot P^T$, where $P$ is a unitary matrix and $Q$ is a diagonal matrix; we will use $(C)^{1/2} = P \cdot Q^{1/2} \cdot P^T$ to denote its factorization such that $C = (C)^{1/2} \cdot (C)^{1/2}$. $\mathrm{Tr}(C)$ represents the trace of a matrix $C$ (sum of diagonals).

## 3. DSI Gaussian Mechanism and Properties

In this section, we present the main algorithm to determine the minimal DSI Gaussian noise and study the interpretation and properties of DSI guarantees. We select $\psi$ as an arbitrary $f$-divergence, while our following results also work for $\psi$ being some non-decreasing function of a $f$-divergence, such as Rényi Divergence. Temporally, we assume a deterministic processing function $\mathcal{F} : \mathcal{U}^* \to \mathbb{R}^d$ whose output is a $d$-dimensional real vector; the generalization to a randomized function can be found in Appendix D. In the following, we simplify the notations of the Gaussian noise, its covariance and the $m$ references by $e$, $\Sigma$ and $\mathsf{R}_{[1:m]}$, respectively, which inherently can be dependent on the input U. We will also use $\mathcal{M}(\cdot) = \mathcal{F}(\cdot) + e$ to denote the noisy version of $\mathcal{F}$.

Let $z_i = \mathcal{F}(\mathsf{R}_i) - \mathcal{F}(\mathsf{U})$ be the output difference between applying the processing function $\mathcal{F}(\cdot)$ on U and $\mathsf{R}_i$. For any given $f$-divergence $\mathcal{D}_f$ and a Gaussian noise $e \sim \mathcal{N}(0, \Sigma)$, the objective divergence in (2) can be expressed as

$$\mathcal{D}_f(\mathbb{P}_{\mathcal{M}(\mathsf{U})} \| \mathbb{P}_{\mathcal{M}(\mathsf{R}_i)}) = \mathcal{D}_f\big(\mathcal{N}(\mathcal{F}(\mathsf{U}), \Sigma) \| \mathcal{N}(\mathcal{F}(\mathsf{R}_i), \Sigma)\big)$$
$$= \mathcal{D}_f\big(\mathcal{N}(0, \Sigma) \| \mathcal{N}(z_i, \Sigma)\big).$$

Here, we utilize the shift-invariance property of $f$-divergence (Lemma B.2). Thus, mathematically, determining the optimal $d$-dimensional Gaussian noise in terms of minimal expected $l_2$-norm square $\mathbb{E}[\|e\|_2^2]$ can be framed as the following optimization with constraints.

$$\min_{\Sigma} \quad \mathbb{E}_{e \sim \mathcal{N}(0,\Sigma)}[\|e\|_2^2] = \min_{\Sigma} \quad \mathrm{Tr}(\Sigma) \tag{5}$$

$$\text{s.t.} \quad \mathcal{D}_f\big(\mathcal{N}(0, \Sigma) \| \mathcal{N}(z_i, \Sigma)\big) \le \gamma_i, i = 1, 2, \cdots, m. \tag{6}$$

In (5), it is noted that the expected $l_2$-norm squared of the noise $e$ corresponds to the trace of its covariance matrix $\Sigma$. As for the constraints (6), we show the following results as a universal simplification.

**Lemma 3.1** ($f$-Divergence between Gaussians). *For an arbitrary $f$-divergence and two (multivariate) Gaussian distributions with identical covariance $\mathcal{N}(a, \Sigma)$ and $\mathcal{N}(b, \Sigma)$, there exists some non-decreasing function $\mathcal{H}_f(\cdot)$ such that*

$$\mathcal{D}_f\big(\mathcal{N}(a, \Sigma) \| \mathcal{N}(b, \Sigma)\big) = \mathcal{H}_f(\|a - b\|_{\Sigma^{-1}}^2). \tag{7}$$

*Here,* $\|t\|_{\Sigma^{-1}}^2 = t\Sigma^{-1}t^T$ *represents the Mahalanobis norm.*

Lemma 3.1 implies that any $f$-divergence between two Gaussians with identical covariance can be expressed as some monotone function of the Mahalanobis norm with respect to the difference of their mean. By Lemma 3.1, let $\mathcal{H}_f^{-1}$ be the inverse function of $\mathcal{H}_f$ and $\tilde{\gamma}_i = \mathcal{H}_f^{-1}(\gamma_i)$, which is uniquely determined, (6) can be rewritten as

$$\|\mathcal{F}(\mathsf{U}) - \mathcal{F}(S_i)\|_{\Sigma^{-1}}^2 = \|z_i\|_{\Sigma^{-1}}^2 \le \tilde{\gamma}_i, \ i \in [1 : m]. \tag{8}$$

Now, to address (5) under the constraints (8), a straightforward approach might involve directly optimizing the $d^2$ parameters of the noise covariance matrix $\Sigma$. However, this approach faces at least two significant challenges:

First, an implicit constraint in (5) requires that $\Sigma$, as a covariance matrix, must be positive semi-definite. Notably, the set of all positive semi-definite matrices is convex but not continuous, making the implementation of projection-based optimization non-trivial. Second, in high-dimensional settings—the primary focus of this paper—the dimensionality $d$, which can reach billions in practical applications, is often much larger than the number of references $m$. Optimizing $d^2$ parameters under such conditions becomes computationally prohibitive. To overcome these challenges, we reformulate the objective problem into an alternative form.

### 3.1. Algorithm for Optimal Gaussian Noise

Our proposed algorithm to determine the minimal noise consists of two components: subspace embedding and Lagrangian multiplier gradient descent (GD), which is finally reduced to optimize only $m$ parameters.

**Subspace Embedding**: In the high-dimensional case when $m \leq d$, it is noted that we only need to calibrate the noise $e$ within the subspace spanned by $z_{[1:m]}$ to obfuscate each difference $z_i$ rather than the entire output space $\mathbb{R}^d$. More importantly, this subspace is of rank up to $m$, and even in the worst case when $z_{[1:m]}$ are all orthogonal to each other, the required noise will only scale in a rate $O(\sqrt{\min\{m,d\}})$ which fundamentally avoids the curse of dimensionality in III mentioned earlier. Therefore, in the high-dimensional case, we only need to determine the noise form in this subspace and afterwards project it back to $\mathbb{R}^d$.

In the following, we use $\mathcal{Z} \in \mathbb{R}^{m \times d}$ to represent the matrix form of $z_{[1:m]}$, where each row corresponds to a $z_i$, and we assume the rank of $\mathcal{Z}$ is $\bar{m}$ with $\bar{m} \leq m \leq d$.

**Lemma 3.2** (Subspace Basis)**.** *Consider the SVD of $\mathcal{Z} \cdot \mathcal{Z}^T$,*

$$\mathcal{Z} \cdot \mathcal{Z}^T = P_{\mathcal{Z}} \cdot Q_{\mathcal{Z}} \cdot P_{\mathcal{Z}}^T, \quad (9)$$

*where $P_{\mathcal{Z}}$ is a unitary matrix whose columns correspond to eigenvectors and $Q_{\mathcal{Z}}$ is diagonal of eigenvalues. Let $\bar{Q}_{\mathcal{Z}} \in \mathbb{R}^{\bar{m} \times \bar{m}}$ and $\bar{P}_{\mathcal{Z}} \in \mathbb{R}^{d \times \bar{m}}$ be the submatrix of $Q_{\mathcal{Z}}$ and $P_{\mathcal{Z}}$ after excluding zero eigenvalues with its eigenvectors. Then, the rows of $\Pi = \bar{Q}_{\mathcal{Z}}^{-1/2} \bar{P}_{\mathcal{Z}}^T \cdot \mathcal{Z}$ form an orthogonal unit basis of the subspace spanned by the rows of $\mathcal{Z}$.*

Lemma 3.2 shows operationally how to find a basis $\Pi$ of the target subspace spanned by $z_{[1:m]}$. It is noted that each row of $\mathcal{X} = \bar{P}_{\mathcal{Z}} \cdot \bar{Q}_{\mathcal{Z}}^{1/2}$ now represents the expression of $\mathcal{Z}$ under the selected basis $\Pi$. In addition, for any $\bar{m}$-dimensional Gaussian noise $\bar{e} \sim \mathcal{N}(0, \bar{\Sigma})$ for $\bar{\Sigma} \in \mathbb{R}^{\bar{m} \times \bar{m}}$, $\mathbb{E}_{\bar{e}} \|\bar{e}\|_2^2 = \mathbb{E}_{\bar{e}} \|\bar{e} \cdot \Pi\|_2^2$, since $\text{Tr}(\bar{\Sigma}) = \text{Tr}((\Pi \cdot \Pi^T) \cdot \bar{\Sigma}) = \text{Tr}(\Pi^T \bar{\Sigma} \cdot \Pi)$. Therefore, the entire problem can be reduced to finding a $\bar{m} \times \bar{m}$ covariance matrix $\bar{\Sigma}$ to

$$\min_{\bar{\Sigma}} \text{Tr}(\bar{\Sigma}), \ s.t. \ \|x_i\|_{\bar{\Sigma}^{-1}}^2 \leq \tilde{\gamma}_i, \ i = 1, 2, \cdots, m. \quad (10)$$

Here, $x_i$ is the $i$-th row of $\mathcal{X}$, the expression of $z_i$ under

---

**Algorithm 1** Optimal DSI Gaussian Mechanism

**Input:** Output differences $\{z_1, z_2, \cdots, z_m\}$ with its matrix form $\mathcal{Z}$ by rows, error threshold $\kappa$, security parameters $\tilde{\gamma}_{[1:m]}$, initialized $\boldsymbol{\lambda} = \lambda_{[1:m]}$ and step size $\eta$.

1: **if** $m \leq d$ **then**
2:     Determine subspace basis $\Pi$ and the expression $\mathcal{X}$ of $\mathcal{Z}$ under $\Pi$ by Lemma 3.2. $\triangleright$ {subspace embedding}
3: **else**
4:     $\Pi = \boldsymbol{I}_{d \times d}$ and $\mathcal{X} = \mathcal{Z}$.
5: **end if**
6: **while** true **do**
7:     $\boldsymbol{\lambda} \leftarrow \max\{\boldsymbol{\lambda} - \eta \cdot \nabla\mathcal{L}(\boldsymbol{\lambda}), \boldsymbol{0}\}$ with gradient $\nabla\mathcal{L}(\boldsymbol{\lambda})$ from (14). $\triangleright$ {GD with projection onto $\mathbb{R}_{\geq 0}^m$}
8:     Compute $\bar{\Sigma}(\boldsymbol{\lambda}) = \left( \sum_{i=1}^m \lambda_i \cdot x_i^T \cdot x_i \right)^{1/2}$.
9:     **if** $\|\nabla_{\mathbb{R}_{\geq 0}^m} \mathcal{L}(\boldsymbol{\lambda})\| \leq \kappa$ for $i = 1, 2, \cdots, m$ **then**
10:       **break**
11:     **end if**
12: **end while**
13: Sample $\bar{e} \sim \mathcal{N}(0, \bar{\Sigma}(\boldsymbol{\lambda}))$ and project it back to $\mathbb{R}^d$ as $e = \bar{e} \cdot \Pi$.

    **Output**: $e \in \mathbb{R}^d$.

---

basis $\Pi$. With the optimum $\bar{\Sigma}^*$ in (10), we can sample $\bar{e} \sim \mathcal{N}(0, \bar{\Sigma}^*) \in \mathbb{R}^{\bar{m}}$ and project it back to $\mathbb{R}^d$ by $\bar{e} \cdot \Pi$.

As a final remark, in the cases when $\bar{m} > d$, there is no need to implement above-mentioned subspace embedding, which is equivalent to $\Pi = \boldsymbol{I}_{d \times d}$, the $d \times d$ identity matrix, and $\mathcal{X} = \mathcal{Z}$, while all following analysis still works.

**Lagrangian Multiplier GD**: We consider the equivalent mix-max dual problem of (10) with Lagrange multipliers $\boldsymbol{\lambda} = \lambda_{[1:m]}, \lambda_i \geq 0$,

$$\min_{\bar{\Sigma}} \max_{\boldsymbol{\lambda} = \lambda_{[1:m]} \geq 0} \text{Tr}(\bar{\Sigma}) + \sum_{i=1}^m \lambda_i(\|x_i\|_{\bar{\Sigma}^{-1}}^2 - \tilde{\gamma}_i). \quad (11)$$

It is not hard to verify that (11) is convex regarding $\bar{\Sigma}$ while linear (concave) with respect to $\boldsymbol{\lambda}$, implying that the optimum $(\bar{\Sigma}^*, \boldsymbol{\lambda}^*)$ of (11) is unique. The following theorem presents further insights that the optimum $\bar{\Sigma}^*$ can be expressed by the Lagrange multipliers $\lambda_{[1:m]}$, and $\mathcal{X}$ and the optimization over $\bar{\Sigma}$ is thereby reduced to determining $\boldsymbol{\lambda}^*$.

**Theorem 3.3** (Optimal Covariance from Optimal Lagrange Multipliers)**.** *Given a selection of $\boldsymbol{\lambda} = \lambda_{[1:m]}$, define*

$$\bar{\Sigma}(\boldsymbol{\lambda}) = \left( \sum_{i=1}^m \lambda_i \cdot x_i^T \cdot x_i \right)^{1/2}. \quad (12)$$

*The optimal solution $\bar{\Sigma}^*$ to (10) is unique in a form $\bar{\Sigma}^* = \bar{\Sigma}(\boldsymbol{\lambda}^*)$ where $\boldsymbol{\lambda}^* = \arg\min_{\boldsymbol{\lambda} \geq 0} \mathcal{L}(\boldsymbol{\lambda})$ is the unique optimum of the function $\mathcal{L}(\boldsymbol{\lambda})$, defined below, under nonnegative constraints $\boldsymbol{\lambda} \geq 0$,*

$$\mathcal{L}(\boldsymbol{\lambda}) = -\left( \text{Tr}(\bar{\Sigma}(\boldsymbol{\lambda})) + \sum_{i=1}^m \lambda_i(\|x_i\|_{\bar{\Sigma}(\boldsymbol{\lambda})^{-1}}^2 - \tilde{\gamma}_i) \right). \quad (13)$$

*Moreover, the gradient of $\mathcal{L}(\boldsymbol{\lambda})$ in (13) has a closed form*

$$\frac{\partial \mathcal{L}(\boldsymbol{\lambda})}{\partial \lambda_i} = -x_i \bar{\Sigma}(\boldsymbol{\lambda})^{-1} x_i^T + \tilde{\gamma}_i. \tag{14}$$

From Theorem 3.3, the entire problem is reduced to a convex optimization (13) over $m$ parameters $\boldsymbol{\lambda}$ under a simple non-negative constraints, which is solvable using standard projected gradient descent (PGD). We summarize this optimal Gaussian mechanism as Algorithm 1 and we discuss additional optimization trick in Appendix C.4.

### 3.2. Interpretation and Properties of DSI Guarantees

**(i) One-Sided v.s. Two-Sided**: Though our Gaussian mechanism ensures a symmetric DSI guarantee, we need to stress that this symmetry does not hold in general, i.e., $\mathcal{D}_f(\mathbb{P}_{\mathcal{M}(\mathsf{U})} \| \mathbb{P}_{\mathcal{M}(\mathsf{R}_i)}) \neq \mathcal{D}_f(\mathbb{P}_{\mathcal{M}(\mathsf{R}_i)} \| \mathbb{P}_{\mathcal{M}(\mathsf{U})})$. $f$-divergence technically is not a symmetric metric between two distributions. But one-sided DSI $\mathcal{D}_f(\mathbb{P}_{\mathcal{M}(\mathsf{U})} \| \mathbb{P}_{\mathcal{M}(\mathsf{R}_i)})$ is usually sufficient to provide desired sematic interpretation. For example, recall the memorization or the backdoor applications discussed in Introduction section:

$R_i$ represents $\mathsf{U}$ excluding some target samples to mitigate memorizing or data from malicious sources; if the reference model $\mathcal{M}(\mathsf{R}_i)$ trained on $\mathsf{R}_i$ is of a high success rate to be robust against either memorization or backdoor attacks, then the one-sided DSI $\mathcal{D}_f(\mathbb{P}_{\mathcal{M}(\mathsf{U})} \| \mathbb{P}_{\mathcal{M}(\mathsf{R}_i)})$ can be translated into an upper bound of the success rate for $\mathcal{M}(\mathsf{U})$, following results such as Lemma 2.3. However, such an one-sided measurement on the reference models in DSI is conceptually different from III guarantees, such as DP, where the measurement must be symmetric; otherwise the asymmetry itself could incur leakage.

**(ii) Postprocessing Immunity**: The processing inequality (Lemma B.1) of $f$-divergence also straightforwardly implies that DSI guarantees are robust to any post-processing. One concrete example is that imagine the processing function $\mathcal{F}$ is some deep learning algorithm and if the noisy model weights satisfies (2), so do its the inference results produced.

**(iii) Composition**: There are many practical applications where we may need to sequentially implement numerous, possibly-adaptive processing functions, where the output from the preceding function can be the input for the subsequent one; and we need to keep track of the cumulative divergence across multiple outputs generated. To be specific, imagine two processing functions $\mathcal{F}_1$ and $\mathcal{F}_2$; in the context of DSI noise mechanism, our goal is to determine two noises $e_1$ and $e_2$ such that for the joint mechanism $\tilde{\mathcal{M}}$

$$\tilde{\mathcal{M}}(\mathsf{U}) = \big(\mathcal{F}_1(\mathsf{U}) + e_1, \mathcal{F}_2(\mathsf{U}, \mathcal{F}_1(\mathsf{U}) + e_1) + e_2\big), \tag{15}$$

the divergence $\mathcal{D}_f(\mathbb{P}_{\bar{\mathcal{M}}(\mathsf{U})} \| \mathbb{P}_{\bar{\mathcal{M}}(\mathsf{R}_i)}) \leq \gamma_i$ for each pair $(\mathsf{U}, \mathsf{R}_i), i \in [1 : m]$. In such an adaptive setup, the input of $\mathcal{F}_2$ includes both $\mathsf{U}$ and the noisy output from previously-

---

**Algorithm 2** DSI Deep Learning Framework

**Input:** a (black-box) optimizer $\mathcal{O}(w, \mathsf{B})$ starting from model weight $w$ with data batch $\mathsf{B}$, an input set $\mathsf{U} = \{u_{[1:n]}\}$ of $n$ datapoints, $m$ reference subsets $\mathsf{R}_i \subset \mathsf{U}$, round number $T$, $T$ batches of samples $\mathsf{B}^{(1)}, \cdots, \mathsf{B}^{(T)}$, $\mathsf{B}^{(t)} \subset \mathsf{U}$ for $t \in [1 : T]$, security budget $\epsilon_i$ for $i \in [1 : m]$, and initialization $w^{(0)}$.

1: With a composition accounting (Theorem 3.4), determine per-round security budget $\epsilon_i^{(t)}$ for $t \in [1 : T]$ given a global budget $\epsilon_i$ for each $i \in [1 : m]$ in (2).
2: **for** $t = 1, 2, \cdots, T$ **do**
3:     Determine $\mathsf{R}_i^{(t)} = \mathsf{R}_i \cap \mathcal{B}^{(t)}$ for $i \in [1 : m]$.
4:     For each $i \in [1 : m]$, compute the outcome difference

$$z_i^{(t)} = \mathcal{O}(w^{(t-1)}, \mathsf{R}_i^{(t)}) - \mathcal{O}(w^{(t-1)}, \mathsf{B}^{(t)}). \tag{16}$$

5:     Apply Algorithm 1 on $z_{[1:m]}^{(t)}$ with budgets $\tilde{\epsilon}_{[1:m]}^{(t)}$ to determine the noise $e^{(t)}$.
6:     Update weights $w^{(t)} = \mathcal{O}(w^{(t-1)}, \mathsf{B}^{(t)}) + e^{(t)}$.
7: **end for**
  **Output**: $w^{(T)}$

---

implemented $\mathcal{F}_1$, and it becomes complicated to even analyze the output distribution from $\bar{\mathcal{M}}$ due to infinite possible instances of $e_1$. Fortunately, we show an sequential implementation of Algorithm 1, iteratively *conditional* on the noise sampled for preceding processing functions, is sufficient to produce *tight* composite DSI bound.

**Theorem 3.4** (Adaptive Composition under $\alpha$-Divergence). *Let $\mathsf{D}_\alpha$ be (non-normalized) $\alpha$-divergence with $f(t) = t^\alpha$ in Definition 2.1. Suppose $e_1$ is produced by Algorithm 1 on $\mathcal{F}_1$ such that $\mathcal{D}_\alpha\big(\mathbb{P}_{\mathcal{F}_1(\mathsf{U})+e_1} \| \mathbb{P}_{\mathcal{F}_1(\mathsf{R}_i)+e_1}\big) \leq \gamma_i^{(1)}$ for each pair $(\mathsf{U}, \mathsf{R}_i)$, $i \in [1 : m]$, and conditional on $e_1$, we implement Algorithm 1 on $\mathcal{F}_2(\cdot, \mathcal{F}_1(\cdot) + e_1)$ which returns $e_2$ such that $\mathcal{D}_\alpha\big(\mathbb{P}_{\mathcal{F}_2(\mathsf{U}, \mathcal{F}_1(\mathsf{U})+e_1)} \| \mathbb{P}_{\mathcal{F}_2(\mathsf{R}_i, \mathcal{F}_1(\mathsf{R}_i)+e_1)}\big) \leq \gamma_i^{(2)}$, then for the joint mechanism $\tilde{M}$ defined in (15),*

$$\mathcal{D}_\alpha\big(\mathbb{P}_{\tilde{\mathcal{M}}(\mathsf{U})} \| \mathbb{P}_{\tilde{\mathcal{M}}(\mathsf{R}_i)}\big) \leq \gamma_i^{(1)} \cdot \gamma_i^{(2)}. \tag{17}$$

*The equality of (17) holds when the noises $e_1$ and $e_2$ are tight to locally produce $\gamma_i^{(1)}$ and $\gamma_i^{(2)}$ for $\mathcal{F}_1$ and $\mathcal{F}_2$, respectively.*

Theorem 3.4 shows that under $\alpha$-divergence, the adaptive composition can be simply expressed as a product of local DSI parameters via (17), which can be easily generalized to arbitrary $T$ adaptive processing functions $\mathcal{F}_{[1:T]}$. More importantly by Theorem 3.4, operationally determining the noises $e_{[1:T]}$ only requires sequentially implementing the adaptive processing functions $\mathcal{F}_{[1:T]}$ *once*, where each $\mathcal{F}_i$ is conditional on earlier noisy outputs. The generalization to other $f$-divergence can be found in Appendix C.5.

**(iv) Group DSI**: Inspired by group privacy (Dwork et al., 2006b), a natural question is given DSI guarantees regarding

each *individual* reference $R_i$, what can we say about the DSI regarding an aggregation of a group $\{R_i\}$? The following lemma upper bounds the group DSI.

**Lemma 3.5** (Group DSI). *For a Gaussian noise $e \sim \mathcal{N}(0, \Sigma)$ such that $\|\mathcal{F}(U) - \mathcal{F}(R_i)\|_{\Sigma^{-1}}^2 \leq \tilde{\gamma}_i$, $i \in [1 : m]$, arbitrary non-negative weights $w_{[1:m]} \in \mathbb{R}_{\geq 0}^m$ such that $\sum_{i=1}^m w_i = 1$, and an subset $\Omega \subset [1 : m]$,*

$$\mathcal{D}_f(\mathbb{P}_{\mathcal{F}(U)+e} \| \mathbb{P}_{\sum_{i \in \Omega} w_i \mathcal{F}(S_i)+e}) \leq \mathcal{H}_f(\sum_{i \in \Omega} w_i^2 \cdot \sum_{i \in \Omega} \tilde{\gamma}_i).$$

Lemma 3.5 is helpful when our target data feature is included in multiple reference set $R_i$. For example, if a piece of information is distributed in multiple references $R_i$ while we only ensure DSI guarantees for each preselected pair $(U, R_i)$ during the training procedure, Lemma 3.5 can be used to provide an aggregated upper bound of memorization risk as discussed later in Section 5.

## 4. Deep Learning with DSI

In this section, we propose a *gray-box* framework of trustworthy deep learning with DSI. It is worth noting that all presented techniques so far do not assume any specific assumptions on the processing function $\mathcal{F}$. Inspired by the iterative perturbation with composition accounting in the well-known DP-SGD (Abadi et al., 2016)[3], we propose a more general framework to support the application of *black-box* optimizer with DSI noise for sharpened trustworthiness-utility tradeoff as Algorithm 2.

In Algorithm 2, we consider a common setup where each $R_i$ is a subset of the input $U$, consisting of $n$ samples. The entire training procedure is divided into $T$ rounds, each employing a black-box optimizer $\mathcal{O}$. In the $t$-th round, a batch $B^{(t)}$ is selected from $U$ as the universe of samples for that round. For each $R_i$, we apply $\mathcal{O}$ starting from the previous iterate $w^{(t-1)}$ on the intersection $B^{(t)} \cap R_i$. We then compute the difference $z_i^{(t)}$ between the updated weights $\mathcal{O}(w^{(t-1)}, B^{(t)} \cap R_i)$ and $\mathcal{O}(w^{(t-1)}, B^{(t)})$, the latter being the update from the full universe $U$. Using composition (Theorem 3.4), we then apply the DSI noise mechanism (Algorithm 1) to determine the per-round noise $e^{(t)}$ required to obfuscate the target updated model weight $\mathcal{O}(w^{(t-1)}, B^{(t)})$.

When we select $R_i$ to be a *leaving-one* subset of $U$ after excluding the $i$-th datapoint and $\mathcal{O}$ as an *one-iteration, per-sample-clipped* gradient descent, Algorithm 2 reduces to regular DP-SGD except we apply optimized DSI noise instead of isotropic DP noise. Compared with DP-SGD, the DSI framework has two major advantages:

1. **Flexible Optimizer with Local Iterations Tricks**: Unlike DP-SGD, which requires a tractable per-round

---

[3]One can alternatively consider end-to-end DSI guarantee by taking $\mathcal{F}$ as the entire training algorithm, but tight analysis can be computationally expensive, as discussed in Appendix E.1.

*sensitivity*—typically enforced by per-sample gradient clipping which restricts neural network architectures such as BatchNorm and incurs additional bias (Xiao et al., 2023b)– Algorithm 2 imposes no such constraints on the construction of $\mathcal{O}$. As a concrete example, we find that a more efficient choice for $\mathcal{O}$ is a multi-iteration optimizer without clipping, such as local SGD (Stich, 2018). This method not only accelerates convergence but also reduces the composition budget for a smaller cumulative divergence.

2. **Sharpened Noise**: For a global $(\epsilon, \delta)$ DP guarantee over $T$ iterations, DP-SGD is known to require a per-round noise in $l_2$-norm $\tilde{O}(\sqrt{dT}/n\epsilon)$ for $d$ model parameters; as a comparison, in DSI setup with $R_i$ being a leaving-one subset of $U$, the noise scale is improved to $\tilde{O}(\sqrt{T}/\sqrt{n}\epsilon)$, which is $O(\sqrt{d/n})$ smaller and independent of dimensionality $d$. More details are deferred to Appendix E.2.

To provide a clearer comparison with the classic $(\epsilon, \delta)$ DP parameters, all our following experiments adopt the same measurement by transforming the $\gamma$ security parameter measured in $\alpha$-divergence (Theorem 3.4) into Rényi divergence, which further upper bounds $(\epsilon, \delta)$ parameters (Proposition 3 in (Mironov, 2017)). In Table 1, we compare the state-of-the-art results of DP-SGD for the training of CIFAR10 in ResNet-20 (0.3M parameters) and WideResNet-16 (2.6M parameters) *from scratch* with those of Algorithm 2 by selecting the optimizer $\mathcal{O}$ as 20 GD iterations in a small batch size 400 *without* clipping. We observe that DSI local-SGD outperforms DP-SGD in all cases, with more significant advantages in larger models. Moreover, empirically, we also observe that DSI local-SGD benefits from the properly selected larger batch size. The DSI results in Table 1 can be further improved through scaling with more computation cost, similar to scaling DP-SGD (De et al., 2022).

## 5. Experiments

We further show the applications of DSI deep learning for various trust concepts. We specify the experimental setup in Section F. For *all* following experiments, DP-SGD requires prohibitively-large noise or the provable $\epsilon$ by regular isotropic DP noise can be hundreds of times larger than that by DSI noise for the same empirical utility (Appendix F.4). Thus, we omit the comparisons with DP-SGD below.

### 5.1. Memorization in LLM

We study the memorization when fine-tuning GPT2 (Radford et al., 2019) and Open Pre-trained Transformer (OPT) (Zhang et al., 2022) using WikiText-5 dataset (Merity et al., 2016). $R_{[1:m]}$ are selected as leaving-one subsets, targeting mitigating memorization on each sample. Two types of memorization are considered.

**a)** $(v, c)$**-exposure** (Carlini et al., 2019) We insert $v$ copies

| Model | Method $\epsilon$ | $\infty$ | 1 | 2 | 3 | 4 | 5 | 6 | 7 | 8 |
|---|---|---|---|---|---|---|---|---|---|---|
| ResNet-20 | (Xiao et al., 2023b) | | 55.3 | 63.1 | 67.6 | 72.4 | 73.7 | 74.3 | 75.8 | 76.0 |
| | (Yu et al., 2021) | 91.7 | / | 59.7 | / | / | 70.1 | / | / | 74.9 |
| | DSI-Local-SGD | | 57.4 | 71.2 | 73.8 | 78.7 | 79.8 | 83.1 | 83.6 | 83.9 |
| WideResNet-16 | (De et al., 2022) | | 56.8 | 64.9 | 69.2 | 71.9 | 74.1 | 77.0 | 78.8 | 79.5 |
| | (Bao et al., 2023) | 94.6 | 57.2 | 64.6 | / | 70.5 | / | / | / | 79.8 |
| | DSI-Local-SGD | | **63.7** | **76.1** | **80.5** | **82.4** | **84.4** | **86.6** | **86.8** | **87.1** |

*Table 1.* **Test Accuracy** (%) Comparison between standard distinguishability control through per-sample gradient clipping and isotropic noise in DP-SGD (Xiao et al., 2023b; De et al., 2022) and the augmented versions with **additional public data** – (Yu et al., 2021) projects per-sample gradients into a 2000-rank subspace estimated by public gradients and (Bao et al., 2023) conducts mixup between every datapoint and public synthetic data – and DSI-Local-SGD with $\mathcal{O}$ being 20-local-GD-iteration with $R_i$ as a leaving-one subset of CIFAR-10 training data U **from scratch without additional data** across different $\epsilon$ selections with $\delta = 10^{-5}$.

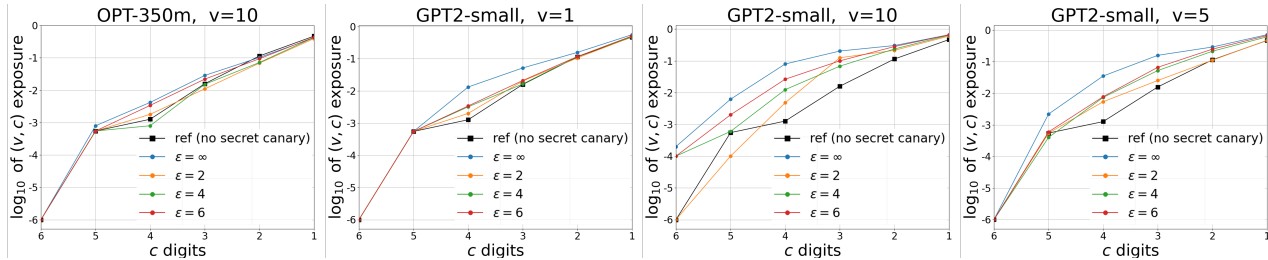

*Figure 2.* $(v, c)$ **Exposure Rate** when fine-tuning GPT2-small and OPT-350M using WikiText5 with/out DSI guarantees $(\epsilon, \delta = 10^{-5})$.

of a *secret canary* with 6 random digits in a form "the wizard's secret code is 572869", into the training data, and consider the probability that the fine-tuned model can recover *at least $c$ out of 6* digits when we prompt with the prefix "the wizard's secret code is" and enforce it to generate 6 digits proportional to the likelihood of digital tokens.

We first approximate the reference probability $P_0(v, c)$ of $(v, c)$-exposure for a model trained on the original dataset *without* secret canaries by a binomial distribution with parameter 0.1: $\sum_{i=c}^{6} \binom{6}{c} \cdot 0.1^c \cdot 0.9^{6-c}$, to guess 6 random digits without prior knowledge. In Fig. 2, we show the exposure rate with a varying duplication number $v$ and different LLMs when protected by different $(\epsilon, \delta)$ DSI guarantees. By Lemma 3.5 and (Dwork et al., 2014), given $(\epsilon, \delta)$ DSI for each leaving-one set $R_i$, $(v, c)$-exposure rate of our model $\mathcal{M}(U)$ is bounded by $e^{v\epsilon} \cdot P_0(v, c) + v e^{(v-1)} \delta$. Comparing with Fig. 2, the theoretical DSI guarantee shows as a conservative bound of the exposure rate. For example in GPT2-small, $(\epsilon = 2, \delta = 10^{-5})$ ensures a provable upper bound of $(v = 1, c = 4)$ exposure by $10^{-2}$ while empirically it is only $2 \cdot 10^{-3}$. In Table 2, we further include the perplexity over validation data as our utility metric, and the *exposure metric* proposed in (Carlini et al., 2019) which approximates the expected ratio (in logarithm $\log_2$) between the likelihood of the fine-tuned model to generate the true 6 digits and that of a random selection of 6 digits.

**b)** $(v, k, c)$-**exact-memorization** (Tirumala et al., 2022): We randomly select 2,000 samples from the training set and duplicate each sample $v$ times. In Table 4 (in Appendix F.2), we report the exact memorization rate with/without DSI. Specifically, we measure the percentage of cases where a

| Duplication($v$), model | $\epsilon = \infty$ | $\epsilon = 2$ | $\epsilon = 4$ | $\epsilon = 6$ |
|---|---|---|---|---|
| 1, GPT2-small | (20.7, 2.2) | (24.3, 0.8) | (22.9, 0.6) | (21.8, 1.5) |
| 5, GPT2-small | (20.5, 11.5) | (24.2, 0.9) | (23.0, 3.2) | (22.0, 5.8) |
| 10, GPT2-small | (20.5, 16.0) | (24.0, 2.4) | (22.8, 3.8) | (21.9, 8.9) |
| 1, OPT-125M | (20.1, 3.3) | (24.7, 0.6) | (22.6, 0.9) | (21.9, 1.1) |
| 5, OPT-125M | (20.2, 11.2) | (24.8, 1.1) | (23.3, 2.8) | (22.3, 2.9) |
| 10, OPT-125M | (20.2, 14.6) | (24.5, 1.9) | (23.0, 3.1) | (22.0, 8.8) |
| 1, OPT-350M | (16.1, 0.6) | (17.6, 0.3) | (16.9, 0.4) | (16.6, 0.5) |
| 10, OPT-350M | (16.1, 8.6) | (17.6, 0.6) | (16.8, 1.7) | (16.5, 5.2) |

*Table 2.* (**Validation Perplexity**, **Exposure Metric** (Carlini et al., 2019)) when fine-tuning different LLMs using WikiText5 with and without ($\epsilon = \infty$) DSI guarantees in $(\epsilon, \delta = 10^{-5})$.

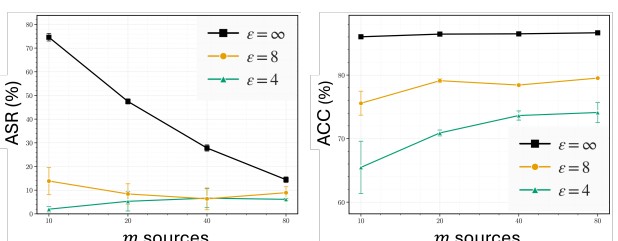

*Figure 3.* **Attack Success Rate** (ASR) and **Test Accuracy** (ACC) when training poisoned CIFAR10 under **Low-Frequency** attacks (Zeng et al., 2021) with/without DSI guarantees $(\epsilon, \delta = 10^{-5})$.

fine-tuned model, when prompted with the first $k = 100$ tokens of each of the 2,000 selected samples, can exactly reproduce the next $c = 5$ tokens. We observe that larger models, requiring more iterations (more exposure to the training data) before convergence, tend to memorize more information. For example, under the same duplication setting of $v = 10$, OPT-350M, which also exhibits lower perplexity, achieves a 39.2% $(10, 100, 5)$ exact memorization rate, compared to 17.9% for OPT-125M, aligning with prior results in (Tirumala et al., 2022). Similarly, provable DSI guaran-

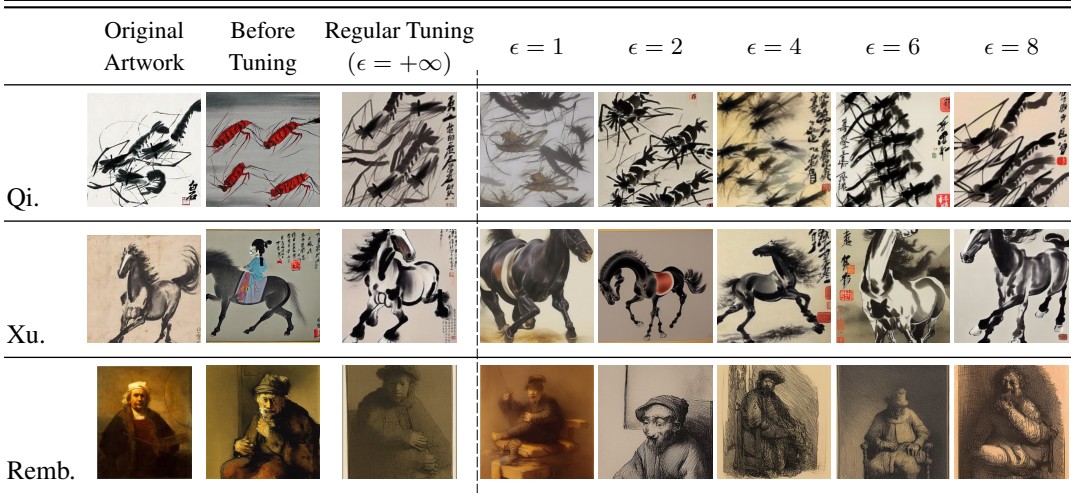

|  | Original Artwork | Before Tuning | Regular Tuning ($\epsilon = +\infty$) | $\epsilon = 1$ | $\epsilon = 2$ | $\epsilon = 4$ | $\epsilon = 6$ | $\epsilon = 8$ |
|---|---|---|---|---|---|---|---|---|
| Qi. | | | | | | | | |
| Xu. | | | | | | | | |
| Remb. | | | | | | | | |

*Table 3.* Finetuing a Stable Diffusion (v1-4) on a collection of paintings from 10 artists with/without DSI under various $\epsilon$ and $\delta = 0.002$.

tees provide conservative upper bounds. For instance, in the case of GPT2-small, we evaluate the same exact memorization on *unseen* test data, which averages 1.66% as our reference. Theoretical privacy bounds with $(\epsilon = 2, \delta = 10^{-5})$ upper limit the $(1, 100, 5)$ exact memorization rate at 12.3%, whereas the empirical rate is only 2.05%.

### 5.2. Data Poisoning (Backdoor) Attacks

We consider two benchmark backdoor attacks: Low-Frequency (Zeng et al., 2021) and Blended (Chen et al., 2017), both designed to mislead the model trained using poisoned data into misclassifying samples containing specific triggers to a target class. We evaluate the DSI defense on poisoned CIFAR-10 dataset in a setting with $m$ data sources, where one source is malicious (a corruption rate of $1/m$). Specifically, we evenly partition the CIFAR-10 training dataset into $m$ subsets, each assigned to one data source. The $(m - 1)$ sources retain clean data, while the malicious source poisons its samples according to the chosen attack strategy. In this DSI setup, U is the union of all samples from the $m$ sources, and each reference set $R_i$ is a subset of U excluding samples from the $i$-th source.

For the processing function, we select $\mathcal{F}$ to be a distributed SGD with a robust gradient aggregation, as detailed in Appendix F.3. For Low-Frequency attacks (Zeng et al., 2021), in Fig. 3 we report both the *Attack Success Rate* (ASR)—the proportion of poisoned samples classified into the target class—and the test accuracy on clean data, with and without DSI guarantees for training PreAct-ResNet18 (Wu et al., 2022). The reported ASR corresponds to the model that achieves the highest clean-data test accuracy across iterations. Similar results are obtained for Blended attacks (Chen et al., 2017) in Fig. 4 (see Appendix F.3). As expected, ASR with DSI defense is significantly lower than that without DSI guarantees, which drops as $m$ increases (a smaller corruption rate $1/m$). On the test accuracy side, a larger $m$

(larger $R_i$ with more overlap) stabilizes the federated learning, leading to less DSI noise with better performance.

### 5.3. Copyright Protection/Contribution Control

We further explore the application of DSI for copyright protection by training a Stable Diffusion model on a dataset U consisting of 425 paintings from 10 artists. Each reference set $R_i$ is selected to be a subset of U excluding one artist's paintings, and the DSI parameter quantifies the per-sample contribution to the resulting diffusion model. We show both the original artworks and the generated images under three conditions: before fine-tuning, after fine-tuning without noise ($\epsilon = \infty$), and after fine-tuning using DSI for three representative artists: Baishi Qi (Qi.), Beihong Xu (Xu.), and Rembrandt van Rijn (Remb.). As expected, a larger $\epsilon$ allows greater influence from the original artwork, resulting in a more faithful style transfer. In practice, users can choose $\epsilon$ for each artwork, which directly governs the degree of influence, and use the influence as the basis for compensation in data usage.

### 6. Conclusion

This paper establishes both the theoretical and algorithmic foundations of DSI, offering a general solution to systematically ensure Differential Trust (DT) across a variety of applications. While our initial results demonstrate a significant improvement in the utility-trust trade-off compared to traditional, such as DP-based, methods, the provable DSI bounds remain conservative. A promising direction to refine the DSI analysis is to incorporate amplification effects from other sources of randomness, such as subsampling (in this paper, we solely rely on randomness introduced by noise injection). Furthermore, it would be intriguing to generalize the notion of "influence" in DT and explore other trustworthiness concepts, such as fairness, from a contribution-oriented perspective.

## Acknowledgment

We would like to thank all anonymous reviewers and area chairs for their insightful and helpful comments.

## Impact Statement

This paper aims to systemically address the trust issues in modern AI applications. We expect the semantic interpretation of the Differential Trust and Data Specific Indistinguishability can also be accessible to the general audience and help policy makers select security parameters. We also believe many other societal concerns beyond memorization, backdoor attacks and copyright mentioned in this paper can be explored, both conceptually and technically, in the presented framework.

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

# A. Related Works

## A.1. Further Comparison between Differential Trust (DT), Differential Privacy (DP) and Individual/Instance-based Differential Privacy

In this subsection, we provide a comprehensive comparison between Differential Trustworthiness (DT) and Differential Privacy (DP), including its variants such as per-sample and individual DP, from three perspectives: motivational, conceptual, and operational.

1. **On Motivation**, as mentioned earlier, both DP and DT leverage indistinguishability to establish probabilistic trust guarantees. Distinguishability measures the closeness of two distributions, which can be interpreted in two ways:

   a) It becomes harder to distinguish the source of a randomly drawn sample.

   b) The behavior of two random variables, $a \sim A$ and $b \sim B$, is similar in terms of probability: $\Pr(a \in O) \approx \Pr(b \in O)$ for any set $O$.

   Privacy (confidentiality protection) typically aims to a) prevent adversaries from recovering secrets from leakage. If the leakages from different secrets are indistinguishable, no useful information is revealed to an adversary.

   In contrast, DT aims to b) use divergence between a target and a set of references to characterize the probability that an algorithm's output satisfies certain trust properties. For a set of reference training datasets $R_1, R_2, \ldots, R_m$, if each trained model $\mathcal{F}(R_i)$ has a 99% probability of not memorizing sensitive information $q_i$, then the divergence between $\mathcal{F}(U)$ and each $\mathcal{F}(R_i)$ provides a measure of the probability that $\mathcal{F}(U)$ does not memorize $q_i$.

2. **Conceptually**, DT justifies the role of data dependence in trust-preserving mechanisms, whereas this remains a challenge in privacy. If a specific sensitive data point $x_0$ becomes a parameter in a privacy guarantee (e.g., achieving $(\epsilon, \delta)$-DP for its membership), this itself leaks information about $x_0$. Section 1.2 discusses why input independence is essential for privacy and why most DP mechanisms need to rely on worst-case sensitivity.

   A key insight from DT is that many trust-related concerns focus on the use (e.g. copyright) or influence (e.g. backdoor) of public data to train a model, or governance (e.g. memorization) to prevent certain behaviors of trained models. For DT applications that do not target information leakage, it suffices to construct indistinguishability with respect to safe reference models. Unlike privacy-preserving operations, this construction does not necessarily require input independence.

   Another distinction is that DT only requires one-way divergence—specifically, the $f$-divergence between the target output $\mathcal{F}(U)$ and reference outputs $\mathcal{F}(R_i)$. In contrast, DP treats two adjacent datasets $X$ and $X'$ symmetrically, requiring controlling two-way divergence to prevent leakage. See Section 3.2(i) for further details. Moreover, the reference set in DT can be arbitrary, not necessarily being leaving-one subsets to capture per-sample influence.

3. **Operationally**, we demonstrate how to determine the optimal anisotropic noise to achieve the required DSI guarantees for black-box processing. However, black-box DP analysis remains open.

   Additionally, since classic DP mechanisms calibrate noise based on the worst case, inlier (average-case) data points may enjoy stronger privacy guarantees—motivating prior work on per-instance (Wang, 2019; Thudi et al., 2024) or individual privacy (Feldman & Zrnic, 2021). However, formally quantifying this average-case amplification using worst-case security parameters remains an open question: isotropic noise cannot provide controllable privacy loss for every specific data point or exploit the average case. Existing works (Wang, 2019; Thudi et al., 2024; Feldman & Zrnic, 2021) can only estimate per-sample/individual privacy loss, while DSI noise shows a way to operationally control any specific instance-based privacy risk.

As a summary, for certain trustworthy concepts related to the differential effect of a single datapoint, such as forgetting individuals, DP has been recognized as providing sufficient (but stronger) guarantees (Golatkar et al., 2020a; Carlini et al., 2021). As discussed earlier, the DSI framework can be viewed as a generalization of the III principle employed in DP, allowing for modeling a broader range of trustworthiness through data-specific references while also achieving significantly sharper trade-offs in utility.

### A.2. Unlearning

An unlearning procedure (Bourtoule et al., 2021) aims to remove the influence of a subset of the training data. Complete unlearning through retrain from scratch is usually computationally expensive, and many approximate solutions are proposed, among which a popular line is to clean the trained model weights based on estimated differential effect from target samples to remove, for example Newton update based on Fisher Information Matrix (Golatkar et al., 2020b) or influence function (Guo et al., 2020). Compared to DSI, existing approximate unlearning generally cannot quantify how much influence from target samples is mitigated and it is also unclear about the application of unlearning in a scenario where we have "forgetting" demands on every data point (Kurmanji et al., 2024).

### A.3. Reference Set

It is also worth noting that the reference sets $R_{[1:m]}$ in DSI is conceptually related to the $k$-NAF copyright framework for generative models (Vyas et al., 2023). The $k$-NAF framework divides the entire training dataset into multiple *disjoint* subsets, ensuring that each copyrighted material appears in only one subset. A model is then trained on each subset as a reference, and a sampling strategy is constructed to ensure that the likelihood of any generated output is close to that of each reference model, say $2^k$ multiplicatively-upper-bounded under maximal divergence. However, in $k$-NAF, the security parameter $k$ is *not* freely adjustable, and the proposed sampling method is largely restricted to generative models. In contrast, the DSI framework is applicable to any black-box processing method and supports arbitrary levels of mitigation for each reference.

## B. Properties of $f$-Divergence

**Lemma B.1** (Data Processing Inequality (Sason & Verdú, 2016)). *Consider a channel that produces $Z$ given $Y$ based on the law described as a conditional distribution $\mathsf{P}_{Z|Y}$. If $\mathsf{P}_Z$ is the distribution of $Z$ when $Y$ is generated by $\mathsf{P}_Y$, and $\mathsf{Q}_Z$ is the distribution of $Z$ when $Y$ is generated by $\mathsf{Q}_Y$, then for any f-divergence $\mathcal{D}_f$,*

$$\mathcal{D}_f(\mathsf{P}_Z\|\mathsf{Q}_Z) \leq \mathcal{D}_f(\mathsf{P}_Y\|\mathsf{Q}_Y).$$

**Lemma B.2** (Shift Invariance (Qiao & Minematsu, 2010)). *For two random variables $A$ and $B$ whose distributions are represented by $\mathbb{P}_A$ and $\mathbb{P}_B$, respectively, for any differentiable and invertible transformation $h$ and any f-divergence $\mathcal{D}_f$,*

$$\mathcal{D}_f\big(\mathbb{P}_A\|\mathbb{P}_B\big) = \mathcal{D}_f\big(\mathbb{P}_{h(A)}\|\mathbb{P}_{h(B)}\big).$$

**Lemma B.3** (Joint Convexity of f-Divergence (Sason & Verdú, 2016)). *For any convex function $f(z)$, $z \cdot f(\frac{z}{v})$ is convex with respect to $(z, v)$. Accordingly, an f-divergence $\mathcal{D}_f(\mathsf{P}\|\mathsf{Q})$ is also jointly convex with respect to $(\mathsf{P}, \mathsf{Q})$.*

## C. Deferred Results of DSI Gaussian Mechanism

### C.1. Proof of Lemma 3.1

With some calculation,

$$
\begin{aligned}
&\mathcal{D}_f(\mathcal{N}(\mu_1, \Sigma)\|\mathcal{N}(\mu_2, \Sigma)) \\
&= c_0 \cdot \int_z f\big(\exp\big(-\frac{1}{2} \cdot (\|z - \mu_1\|_{\Sigma^{-1}}^2 - \|z - \mu_2\|_{\Sigma^{-1}}^2)\big) \cdot \exp(-\frac{1}{2} \cdot \|z - \mu_2\|_{\Sigma^{-1}}^2)\big) dz \\
&= c_0 \cdot \int_z f\big(\exp\big(-\frac{1}{2} \cdot (\|z - (\mu_1 - \mu_2)\|_{\Sigma^{-1}}^2 - \|z\|_{\Sigma^{-1}}^2)\big) \cdot \exp(-\frac{1}{2} \cdot \|z\|_{\Sigma^{-1}}^2)\big) dz.
\end{aligned}
\tag{18}
$$

Here, $c_0$ is the normalization constant of multivariate Gaussian density function. Decompose $\Sigma^{-1} = Q^T \cdot \Lambda^2 \cdot Q$, where $Q \cdot Q^T = \boldsymbol{I}_d$ and $\Lambda$ is a diagonal matrix with positive diagonal elements. Then, by taking $v = \Lambda Q z$ and $\bar{\mu} = \Lambda Q(\mu_1 - \mu_2)$, we have

$$\mathcal{D}_f(\mathcal{N}(\mu_1, \Sigma)\|\mathcal{N}(\mu_2, \Sigma)) = c_0 \cdot \int_v f\big(\exp\big(-\frac{1}{2} \cdot (\|v - \bar{\mu}\|^2 - \|v\|^2)\big) \cdot \exp(-\frac{1}{2} \cdot \|v\|^2)\big) dv. \tag{19}$$

Now, let $A$ be the subspace of $\mathbb{R}^d$ of $(d-1)$-rank orthogonal to $\bar{\mu}$ and we can rewrite an arbitrary $v \in \mathbb{R}^d$ as $v = t\bar{\mu} + v_0$ for some $v_0 \in A$, where $\|v\|^2 = t^2\|\bar{\mu}\|^2 + \|v_0\|^2$, then (19) can be further expanded as

$$
\begin{aligned}
&\mathcal{D}_f(\mathcal{N}(\mu_1,\Sigma)\|\mathcal{N}(\mu_2,\Sigma)) \\
&= c_0 \cdot \int_{t\in\mathbb{R}} \int_{v_0} f\big(\exp\big((t-\tfrac{1}{2})\cdot\|\bar{\mu}\|^2\big)\cdot\exp\big(-\tfrac{t^2}{2}\cdot\|\bar{\mu}\|^2\big)\cdot\exp\big(-\tfrac{1}{2}\cdot\|v_0\|^2\big)dv_0\|\bar{\mu}\|dt \\
&= c_1 \cdot \int_{s\in\mathbb{R}} f\big(\exp\big(\|\bar{\mu}\|(s-\tfrac{1}{2}\cdot\|\bar{\mu}\|)\big)\big)\cdot\exp(-\tfrac{1}{2}s^2)ds.
\end{aligned}
\tag{20}
$$

Here, $s = \|\bar{\mu}\|t$. The remaining proof is straightforward.

Next, we want to show (20) is monotonically increasing in $\|\bar{\mu}\|$. Denote $\|\bar{\mu}\|$ by $\theta$ and $\mathcal{D}_f(\mathcal{N}(\mu_1,\Sigma)\|\mathcal{N}(\mu_2,\Sigma))$ by $\bar{\mathcal{H}}_f(\theta)$, then it suffices to show $\frac{d\bar{\mathcal{H}}_f(\theta)}{d\theta} \geq 0$ whenever $\theta \geq 0$.

$$
\begin{aligned}
\frac{d\bar{\mathcal{H}}_f(\theta)}{d\theta} &= c_1 \int_{s\in\mathbb{R}} f'\big(\exp\big(\theta s - \tfrac{1}{2}\theta^2\big)\big)\cdot\exp\big(\theta s - \tfrac{1}{2}\theta^2\big)\cdot(s-\theta)\cdot\exp(-\tfrac{1}{2}s^2)ds \\
&= c_1 \int_{s\in\mathbb{R}} f'\big(\exp\big(\theta s - \tfrac{1}{2}\theta^2\big)\big)\cdot(s-\theta)\exp\big(-\tfrac{1}{2}(s-\theta)^2\big)ds \\
&\xLeftrightarrow{x:=s-\theta} c_1 \int_{x\in\mathbb{R}} f'\big(\exp\big(\theta x + \tfrac{1}{2}\theta^2\big)\big)\cdot x\exp\big(-\tfrac{1}{2}x^2\big)dx \\
&\xLeftrightarrow{\text{denote } f'(\exp(\theta x + \frac{1}{2}\theta^2)) \text{ by } \Phi(x)} c_1 \int_0^\infty \big(\Phi(x)-\Phi(-x)\big)\cdot x\exp\big(-\tfrac{1}{2}x^2\big)dx \geq 0.
\end{aligned}
\tag{21}
$$

Since $f$ is convex, $f'$ is monotonically increasing. We conclude that $\Phi(x) = f'\big(\exp\big(\theta x + \tfrac{1}{2}\theta^2\big)\big)$ is monotonically increasing in $x$ given $\theta \geq 0$. Then the result is apparent as the integrand is non-negative.

### C.2. Proof of Lemma 3.2

For the given $\Pi = \bar{Q}_{\mathcal{Z}}^{-1/2}\bar{P}_{\mathcal{Z}}^T \cdot \mathcal{Z} \in \mathbb{R}^{\bar{m}\times d}$, it is noted that

$$
\Pi \cdot \Pi^T = \bar{Q}_{\mathcal{Z}}^{-1/2}\bar{P}_{\mathcal{Z}}^T \cdot (\mathcal{Z}\cdot\mathcal{Z}^T)\cdot\bar{P}_{\mathcal{Z}}\bar{Q}_{\mathcal{Z}}^{-1/2} = \bar{Q}_{\mathcal{Z}}^{-1/2}\bar{P}_{\mathcal{Z}}^T\cdot(P_{\mathcal{Z}}Q_{\mathcal{Z}}P_{\mathcal{Z}}^T)\cdot\bar{P}_{\mathcal{Z}}\bar{Q}_{\mathcal{Z}}^{-1/2} = \boldsymbol{I}_{\bar{m}\times\bar{m}}.
\tag{22}
$$

(22) suggests that each row of $\Pi$ is of $l_2$-norm 1 and orthogonal to each other. On the other hand, by the definition, the rows of $\Pi$ are within the subspace spanned by the rows of $\mathcal{Z}$, and thus, as claimed, the rows of $\Pi$ form a orthogonal unit basis.

### C.3. Proof of Theorem 3.3

Recall (11). Since the inverse map $\Sigma \mapsto \Sigma^{-1}$ is operator convex on the cone of positive-definite matrices, applying $x(\cdot)x^T$ to both sides preserves the inequality, yielding $\|x\|_{\Sigma^{-1}}^2$ is convex in $\Sigma \succ 0$. It is clear that $\mathrm{Tr}(\Sigma)$ is convex in $\Sigma \succ 0$, thus (11) is convex regarding $\Sigma$ where $\Sigma \succ 0$ and concave (linear) with respect to $\boldsymbol{\lambda}$. Therefore, by Sion's Minimax Theorem,

$$
\begin{aligned}
(\Sigma^*, \boldsymbol{\lambda}^*) &= \arg\min_{\bar{\Sigma}\succ 0}\max_{\boldsymbol{\lambda}=\lambda_{[1:m]}\geq 0}\mathrm{Tr}(\bar{\Sigma}) + \sum_{i=1}^m \lambda_i(\|x_i\|_{\bar{\Sigma}^{-1}}^2 - \tilde{\gamma}_i) \\
&= \arg\max_{\boldsymbol{\lambda}=\lambda_{[1:m]}\geq 0}\min_{\bar{\Sigma}\succ 0} = \mathrm{Tr}(\bar{\Sigma}) + \sum_{i=1}^m \lambda_i(\|x_i\|_{\bar{\Sigma}^{-1}}^2 - \tilde{\gamma}_i)
\end{aligned}
\tag{23}
$$

for a fixed $\boldsymbol{\lambda}$, and

$$
\begin{aligned}
\frac{\partial\big(\mathrm{Tr}(\bar{\Sigma}) + \sum_{i=1}^m \lambda_i(\|x_i\|_{\bar{\Sigma}^{-1}}^2 - \tilde{\gamma}_i)\big)}{\partial\bar{\Sigma}} &= \boldsymbol{I} + \sum_{i=1}^m \lambda_i(-\bar{\Sigma}^{-1}x_i^T x_i\bar{\Sigma}^{-1}) \\
&= \boldsymbol{I} - \bar{\Sigma}^{-1}\big(\sum_{i=1}^m \lambda_i x_i^T x_i\big)\bar{\Sigma}^{-1}.
\end{aligned}
\tag{24}
$$

Setting (24) equaling 0 yields (12). It is clear that (12) is positive definite for any $\boldsymbol{\lambda} \succ 0$ satisfying $\|\boldsymbol{\lambda}\|_0 \geq \dim x_i$, confirming the optimality of (12) in $\mathcal{S}_{++}^{\dim x_i}$.

We then present the derivation of the closed-form gradient of $\mathcal{L}(\boldsymbol{\lambda})$ in (13). Notice that by (12),

$$\bar{\Sigma}(\boldsymbol{\lambda}) = \bar{\Sigma}(\boldsymbol{\lambda})^2 \bar{\Sigma}(\boldsymbol{\lambda})^{-1} = \big( \sum_{i=1}^{m} \lambda_i x_i^T x_i \big) \bar{\Sigma}(\boldsymbol{\lambda})^{-1} = \sum_{i=1}^{m} \lambda_i x_i^T x_i \bar{\Sigma}(\boldsymbol{\lambda})^{-1} \tag{25}$$

Therefore,

$$\mathrm{Tr}\left(\bar{\Sigma}(\boldsymbol{\lambda})\right) = \sum_{i=1}^{m} \lambda_i \, \mathrm{Tr}\left(x_i^T x_i \bar{\Sigma}(\boldsymbol{\lambda})^{-1}\right) = \sum_{i=1}^{m} \lambda_i x_i \bar{\Sigma}(\boldsymbol{\lambda})^{-1} x_i^T = \sum_{i=1}^{m} \lambda_i \|x_i\|_{\bar{\Sigma}(\boldsymbol{\lambda})}^2;$$

$$\mathcal{L}(\boldsymbol{\lambda}) = -2 \, \mathrm{Tr}\left(\bar{\Sigma}(\boldsymbol{\lambda})\right) + \sum_{i=1}^{m} \lambda_i \tilde{\gamma}_i. \tag{26}$$

Finally, for the gradient computation, let $A = \sum_{i=1}^{n} \lambda_i x_i^T x_i$ and by the chain rule

$$\begin{aligned}
\frac{\partial \, \mathrm{Tr}(\bar{\Sigma}(\boldsymbol{\lambda}))}{\partial \lambda_i} &= \left\langle \frac{\partial \, \mathrm{Tr}(\bar{\Sigma}(\boldsymbol{\lambda}))}{\partial A}, \frac{\partial A}{\partial \lambda_i} \right\rangle_F = \big( \frac{1}{2} \cdot A^{-\frac{1}{2}} \big) \cdot \big( x_i^T \cdot x_i \big) \\
&= \frac{1}{2} \cdot \sum_{j,l} \bar{\Sigma}^{-1}(\boldsymbol{\lambda})(j,l) \cdot x_i(k) \cdot x_i(l) = \frac{1}{2} \cdot x_i \bar{\Sigma}^{-1}(\boldsymbol{\lambda}) x_i^T.
\end{aligned} \tag{27}$$

Then (14) follows immediately from (26) and (27).

### C.4. Enhanced Optimization Techniques for Algorithm 1

The projection onto the non-negative orthant introduces non-smoothness at the boundary, which may slow down the convergence in practice. To effectively handle the non-negativity constraint on $\boldsymbol{\lambda}$ and improve convergence efficiency near the boundary, we perform gradient updates in the logarithmic domain. Specifically, we update $\boldsymbol{\lambda}$ as:

$$\boldsymbol{\lambda} := \exp(\log(\boldsymbol{\lambda}) - \eta \nabla \mathcal{L}(\boldsymbol{\lambda})).$$

This formulation aligns with Mirror Descent using a log-barrier function, naturally leading to a multiplicative update that ensures strict positivity of $\boldsymbol{\lambda}$ throughout optimization.

To further enhance convergence speed, we incorporate an adaptive *backtracking line search strategy* for step size $\eta$, combined with periodic resetting:

- **Initialization:** Set the initial step size as $\eta := \eta_0$.

- **Backtracking Line Search:** At each iteration $k$, compute the candidate update:

$$\boldsymbol{\lambda}_k^{\text{new}} = \exp\left(\log \boldsymbol{\lambda}_k - \eta \nabla \mathcal{L}(\boldsymbol{\lambda}_k)\right).$$

  If the function value does not decrease, i.e.,

$$\mathcal{L}(\boldsymbol{\lambda}_k^{\text{new}}) > \mathcal{L}(\boldsymbol{\lambda}_k),$$

  we iteratively reduce $\eta$ using a decay factor $\alpha \in (0,1)$:

$$\eta \leftarrow \alpha \eta.$$

  This process continues until a sufficient decrease is achieved:

$$\mathcal{L}(\boldsymbol{\lambda}_k^{\text{new}}) \leq \mathcal{L}(\boldsymbol{\lambda}_k).$$

- **Periodic Step Size Resetting:** To prevent the step size from becoming excessively small and stalling progress, we periodically reset it to $\eta_0$ every $K_0$ iterations:

$$\eta \leftarrow \eta_0, \quad \text{if } k \mod K_0 = 0.$$

This adaptive search mechanism enables the algorithm to take advantage of large step sizes whenever possible while preventing stagnation due to overly conservative updates. In our implementation, we set $\eta_0 = 10$, $\alpha = 0.1$, and $K_0 = 20$.

## C.5. Composition

### C.5.1. PROOF OF THEOREM 3.4 AND GENERALIZATION

For notation simplicity, we use $\mathcal{M}_1 = \mathcal{F}_1(\cdot) + e_1$ and $\mathcal{M}_2 = \mathcal{F}_2(\cdot, \mathcal{F}_1(\cdot) + e_1) + e_2$ to denote the noisy mechanism of $\mathcal{F}_1$ and $\mathcal{F}_2$, respectively. The $\alpha$-divergence $\mathcal{D}_\alpha\big(\mathbb{P}_{\tilde{\mathcal{M}}(\mathsf{U})} \| \mathbb{P}_{\tilde{\mathcal{M}}(\mathsf{R}_i)}\big)$, can be expanded and rewritten as

$$
\begin{aligned}
\mathcal{D}_\alpha\big(\mathbb{P}_{\tilde{\mathcal{M}}(\mathsf{U})} \| \mathbb{P}_{\tilde{\mathcal{M}}(\mathsf{R}_i)}\big) &= \int_{o^{(1)}, o^{(2)}} \frac{\Pr\big(\bar{\mathcal{M}}(\mathsf{U}) = (o^{(1)}, o^{(2)})\big)^\alpha}{\Pr\big(\bar{\mathcal{M}}(\mathsf{R}_i) = (o^{(1)}, o^{(2)})\big)^{\alpha-1}} \\
&= \int_{o^{(1)}, o^{(2)}} \frac{\big(\Pr\big(\mathcal{M}_1(\mathsf{U}) = o^{(1)}\big) \cdot \Pr\big(\mathcal{M}_2(\mathsf{U}, o^{(1)}) = o^{(2)}\big)\big)^\alpha}{\big(\Pr\big(\mathcal{M}_1(\mathsf{R}_i) = o^{(1)}\big) \cdot \Pr\big(\mathcal{M}_2(\mathsf{R}_i, o^{(1)}) = o^{(2)}\big)\big)^{\alpha-1}} \\
&= \int_{o^{(1)}} \frac{\Pr\big(\mathcal{M}_1(\mathsf{U}) = o^{(1)}\big)^\alpha}{\Pr\big(\mathcal{M}_1(\mathsf{R}_i) = o^{(1)}\big)^{\alpha-1}} \cdot \int_{o^{(2)}} \frac{\Pr\big(\mathcal{M}_2(\mathsf{U}, o^{(1)}) = o^{(2)}\big)^\alpha}{\Pr\big(\mathcal{M}_2(\mathsf{R}_i, o^{(1)}) = o^{(2)}\big)^{\alpha-1}} \\
&= \int_{o^{(1)}} \frac{\Pr\big(\mathcal{M}_1(\mathsf{U}) = o^{(1)}\big)^\alpha}{\Pr\big(\mathcal{M}_1(\mathsf{R}_i) = o^{(1)}\big)^{\alpha-1}} \cdot \mathcal{D}_\alpha\big(\mathbb{P}_{\tilde{\mathcal{M}}_2(\mathsf{U}, o^{(1)})} \| \mathbb{P}_{\tilde{\mathcal{M}}_2(\mathsf{R}_i, o^{(1)})}\big).
\end{aligned}
\tag{28}
$$

By the assumption of the iterative implementation of the noise mechanism Algorithm 1, which always ensures that conditional on any outcome $o^{(1)}$ from $\mathcal{M}_1$, $\mathcal{D}_\alpha\big(\mathbb{P}_{\tilde{\mathcal{M}}_2(\mathsf{U}, o^{(1)})} \| \mathbb{P}_{\tilde{\mathcal{M}}_2(\mathsf{R}_i, o^{(1)})}\big) \leq \gamma^{(2)}$, (28) is further bounded by

$$
\mathcal{D}_\alpha\big(\mathbb{P}_{\tilde{\mathcal{M}}(\mathsf{U})} \| \mathbb{P}_{\tilde{\mathcal{M}}(\mathsf{R}_i)}\big) \leq \int_{o^{(1)}} \frac{\Pr\big(\mathcal{M}_1(\mathsf{U}) = o^{(1)}\big)^\alpha}{\Pr\big(\mathcal{M}_1(\mathsf{R}_i) = o^{(1)}\big)^{\alpha-1}} \cdot \gamma^{(2)} \leq \mathcal{D}_\alpha\big(\mathbb{P}_{\mathcal{M}_1(\mathsf{U})} \| \mathbb{P}_{\mathcal{M}_1(\mathsf{R}_i)}\big) \cdot \gamma^{(2)} \leq \gamma^{(1)} \cdot \gamma^{(2)},
\tag{29}
$$

which completes the proof. It is worthwhile noting that the equality in (29) holds when the noise $e_1$ and $e_2$ tightly produce the required security parameters $\gamma^{(1)}$ and $\gamma^{(2)}$, i.e., $\mathcal{D}_\alpha\big(\mathbb{P}_{\mathcal{M}_1(\mathsf{U})} \| \mathbb{P}_{\mathcal{M}_1(\mathsf{R}_i)}\big) = \gamma^{(1)}$ and $\mathcal{D}_\alpha\big(\mathbb{P}_{\mathcal{M}_2(\mathsf{U}, o^{(1)})} \| \mathbb{P}_{\mathcal{M}_2(\mathsf{R}_i, o^{(1)})}\big) = \gamma^{(2)}$, i.e., the per-round noise computed is optimal.

With a similar reasoning, by selecting $\mathcal{D}_f$ to be KL divergence, we have the following corollary.

**Corollary C.1** (Adaptive DSI Composition under KL-Divergence). *With the same setup, by replacing the $\alpha$-divergence $\mathcal{D}_\alpha$ in Theorem 3.4 with KL divergence $\mathcal{D}_{KL}$,*

$$
\mathcal{D}_{KL}\big(\mathbb{P}_{\bar{\mathcal{M}}(\mathsf{U})} \| \mathbb{P}_{\bar{\mathcal{M}}(\mathsf{R}_i)}\big) \leq \gamma_i^{(1)} + \gamma_i^{(2)}.
\tag{30}
$$

*Proof.*

$$
\begin{aligned}
&\mathcal{D}_{KL}\big(\mathbb{P}_{\tilde{\mathcal{M}}(\mathsf{U})} \| \mathbb{P}_{\tilde{\mathcal{M}}(\mathsf{R}_i)}\big) \\
&= \int_{o^{(1)}, o^{(2)}} \Pr\big(\bar{\mathcal{M}}(\mathsf{U}) = (o^{(1)}, o^{(2)})\big) \cdot \log\big(\frac{\Pr\big(\bar{\mathcal{M}}(\mathsf{U}) = (o^{(1)}, o^{(2)})\big)}{\Pr\big(\bar{\mathcal{M}}(\mathsf{R}_i) = (o^{(1)}, o^{(2)})\big)}\big) \\
&= \int_{o^{(1)}, o^{(2)}} \Pr\big(\mathcal{M}_1(\mathsf{U}) = o^{(1)} \cdot \Pr\big(\mathcal{M}_2(\mathsf{U}, o^{(1)}) = o^{(2)}\big)\big) \cdot \log\big(\frac{\Pr\big(\mathcal{M}_1(\mathsf{U}) = o^{(1)}\big) \cdot \Pr\big(\mathcal{M}_2(\mathsf{U}, o^{(1)}) = o^{(2)}\big)}{\Pr\big(\mathcal{M}_1(\mathsf{R}_i) = o^{(1)}\big) \cdot \Pr\big(\mathcal{M}_2(\mathsf{R}_i, o^{(1)}) = o^{(2)}\big)}\big) \\
&= \int_{o^{(1)}, o^{(2)}} \Pr\big(\mathcal{M}_1(\mathsf{U}) = o^{(1)}\big) \Pr\big(\mathcal{M}_2(\mathsf{U}, o^{(1)}) = o^{(2)}\big) \cdot \big(\log\big(\frac{\Pr\big(\mathcal{M}_1(\mathsf{U}) = o^{(1)}\big)}{\Pr\big(\mathcal{M}_1(\mathsf{R}_i) = o^{(1)}\big)}\big) + \log\big(\frac{\Pr\big(\mathcal{M}_2(\mathsf{U}, o^{(1)}) = o^{(2)}\big)}{\Pr\big(\mathcal{M}_2(\mathsf{R}_i, o^{(1)}) = o^{(2)}\big)}\big)\big) \\
&= \mathcal{D}_{KL}\big(\mathbb{P}_{\mathcal{M}_1(\mathsf{U})} \| \mathbb{P}_{\mathcal{M}_1(\mathsf{R}_i)}\big) + \int_{o^{(1)}} \mathcal{D}_{KL}\big(\mathbb{P}_{\mathcal{M}_2(\mathsf{U}, o^{(1)})} \| \mathbb{P}_{\mathcal{M}_2(\mathsf{R}_i, o^{(1)})}\big) \leq \gamma^{(1)} + \gamma^{(2)}.
\end{aligned}
\tag{31}
$$

Similarly, when $e_1$ and $e_2$ are tightly to produce the per-round DSI guarantees, i.e., $\mathcal{D}_{KL}\big(\mathbb{P}_{\mathcal{M}_1(\mathsf{U})} \| \mathbb{P}_{\mathcal{M}_1(\mathsf{R}_i)}\big) = \gamma^{(1)}$ and $\mathcal{D}_{KL}\big(\mathbb{P}_{\mathcal{M}_2(\mathsf{U}, o^{(1)})} \| \mathbb{P}_{\mathcal{M}_2(\mathsf{R}_i, o^{(1)})}\big) = \gamma^{(2)}$, then the equality holds in (31). □

### C.5.2. COMPARISON BETWEEN DSI AND III COMPOSITION

The composition of III guarantees has been extensively studied, particularly in the context of differential privacy (DP) (Kairouz et al., 2015; Mironov, 2017; Dong et al., 2022; Zhu et al., 2022). The primary goal of composition analysis, whether

in the context of Data-Specific Indistinguishability (DSI) or III, is to upper bound the divergence of the joint distribution given the divergence of individual components. However, most existing III results focus on the composition of the worst-case scenario for each component. This is often formalized as follows: given mechanisms $\mathcal{M}_1(\cdot), \mathcal{M}_2(\cdot), \dots, \mathcal{M}_T(\cdot)$, each satisfying $(\epsilon_0, \delta_0)$-DP, the joint mechanism $\mathcal{M}_T \circ \cdots \circ \mathcal{M}_2 \circ \mathcal{M}_1(\cdot)$ satisfies $\tilde{O}(\sqrt{T}\epsilon_0, T\delta_0)$-DP, as stated in the advanced composition theorem for DP (Kairouz et al., 2015).

In practical applications, such as DP-SGD, where each component $\mathcal{M}_i$ corresponds to the noisy, clipped per-sample gradient aggregation in a single iteration, the security parameter $(\epsilon_0, \delta_0)$ is determined only by the clipping threshold and noise scale. These parameters reflect the worst-case divergence between two adjacent datasets. Consequently, this worst-case composition often fails to tightly characterize the composite divergence for specific outputs generated from particular input pairs, since the worst-case scenario is assumed to happen in every iteration, making these bounds overly conservative.

In contrast, the composition of DSI noise mechanisms captures the average-case behavior. Here, the noise is adaptively aligned with each instance during the composition process. As demonstrated, when the noise is tightly calibrated to produce the per-round DSI guarantees for each individual instance, the resulting composition bound is also tight for the overall randomized joint mechanism.

### C.6. Proof of Lemma 3.5

Based on Lemma 3.1, given $e \sim \mathcal{N}(0, \Sigma)$ such that $\|z_i\|_{\Sigma^{-1}}^2 = \tilde{\gamma}_i$ for $i = 1, 2, \cdots, m$,

$$\mathcal{D}_f(\mathbb{P}_{\mathcal{F}(\mathsf{U})+e} \| \mathbb{P}_{\sum_{i \in \Omega} w_i \mathcal{F}(\mathsf{S}_i)+e}) = \mathcal{H}_f\big(\|\sum_{i \in \Omega} w_i(\mathcal{F}(\mathsf{S}_i) - \mathcal{F}(\mathsf{U}))\|_{\Sigma^{-1}}^2\big) = \mathcal{H}_f\big(\|\sum_{i \in \Omega} w_i z_i\|_{\Sigma^{-1}}^2\big) \tag{32}$$

On the other hand, by Cauchy–Schwarz inequality, with $|\Omega|$ representing the number of elements in $\Omega$.

$$\|\sum_{i \in \Omega} w_i z_i\|_{\Sigma^{-1}}^2 \leq \min\{(\sum_{i \in \Omega} w_i^2)(\sum_{i \in \Omega} \tilde{\gamma}_i), |\Omega| \sum_{i \in \Omega} w_i^2 \tilde{\gamma}_i\}. \tag{33}$$

Thus, combined with the monotone property of $\mathcal{H}_f$, the claim follows.

## D. Generalization to Randomized Processing Function

In Section 3, we mainly focused on the deterministic processing function $\mathcal{F}$ and the entire randomization is from the added Gaussian noise. Our theory can be easily generalized to the randomized $\mathcal{F}(\cdot, \theta)$ with inherent randomness captured by a random seed $\theta$, where $\theta \sim \mathcal{D}_\Theta$. The key idea here is to take the Gaussian noise $e(\theta)$ also dependent on the random seed $\theta$. Suppose for each $\theta$, we determine an $e(\theta) \sim \mathcal{N}\big(0, \Sigma(\theta)\big)$ such that for each difference $z_i(\theta) = \mathcal{F}(\mathsf{R}_i, \theta) - \mathcal{F}(\mathsf{U}, \theta)$, $\mathcal{D}_f\big(\mathcal{N}(0, \Sigma(\theta)) \| \mathcal{N}(z_i(\theta), \Sigma(\theta))\big) \leq \gamma_i$, then provided the joint convexity of f-divergence (Lemma B.3),

$$\mathcal{D}_f(\mathbb{P}_{\mathcal{M}(\mathsf{U})} \| \mathbb{P}_{\mathcal{M}(\mathsf{R}_i)}) = \mathcal{D}_f\big(\sum_\theta \mathbb{P}(\theta)\mathcal{N}(\mathcal{F}(\mathsf{U}, \theta), \Sigma(\theta)) \| \sum_\theta \mathbb{P}(\theta)\mathcal{N}(\mathcal{F}(\mathsf{R}_i, \theta), \Sigma(\theta)))$$
$$\leq \sum_\theta \mathbb{P}(\theta)\mathcal{D}_f\big(\mathcal{N}(0, \Sigma(\theta)) \| \mathcal{N}(z_i(\theta), \Sigma(\theta))\big) \leq \gamma_i. \tag{34}$$

(34) has important operational implication on ensuring DSI for randomized processing procedure. With a similar implementation principle in determining noise in compositional DSI (Theorem 3.4) where there is no need to simulate all possibilities but only focus on the particular sequence of produced instances, for a randomized processing function $\mathcal{F}$, it suffices to 1) randomly sample $\theta$ *only once*, 2) apply algorithm in Section 3.1 to determine the Gaussian distribution $\mathcal{N}\big(0, \Sigma(\theta)\big)$ for the particular difference $z_i(\theta)$, and 3) sample noise $e(\theta) \sim \mathcal{N}\big(0, \Sigma(\theta)\big)$ and output $\mathcal{F}(\mathsf{U}, \theta) + e(\theta)$.

## E. DSI Deep Learning

### E.1. Why not End-to-End DSI for Deep Learning

Another natural question in the construction of the DSI deep learning framework is that, provided the capacity of DSI to handle black-box processing function, can we simply take $\mathcal{F}$ as the entire deep learning algorithm and simply do end-to-end analysis by only adding noise to the last iterate. Unfortunately, at least from computational efficiency standpoint, it remains challenging to tightly determine the optimal or usable noise parameters in such an end-to-end setup.

We implement the following experiment to compare ResNet20 models trained by standard SGD on **adjacent** CIFAR10 dataset. Here, we craft a pair of adjacent datasets $\bar{U}$ and $\bar{U}'$ by considering the original full CIFAR10 set and its subset after excluding the first data point. We first select 100 random seeds, fix them, and train 100 models [4] from each of $\bar{U}$ and $\bar{U}'$ based on the selected 100 seeds with the *same* initialized weight, denoted by $\{w_1, w_2, \cdots, w_{100}\}$ and $\{w'_1, w'_2, \cdots, w'_{100}\}$, respectively. We record the average of the $l_2$-norm of the model weights, where $\sum_{i=1}^{100} \|w_i\|_2/100 = 28.4$ and $\sum_{i=1}^{100} \|w'_i\|_2/100 = 28.2$; on the other hand, we compute their minimal difference $\min_{i,j} \|w_i - w'_j\|_2 = 10.1$. That is to say, across 100 trials, we *fail* to find a single case where SGD on $\bar{U}$ and $\bar{U}'$ converge to two *close* local minimum. As a consequence, if we restrict the randomness of SGD among the 100 selected seeds, the required noise on last-iterate output can be much larger than the model itself and fully destroy the utility.

The above experiments suggest two facts: even if we only drop a single datapoint, a) SGD in deep learning is *not* deterministically robust with fixed random seed; b) its distributional robustness or stability with random seeds *cannot* be computationally verified or exploited. In particular for b), our observation where models trained across 100 trials all converge to very different local minima is not surprising. It is known that the number of local minima actually grows exponentially with the dimension in neural network (Auer et al., 1995). Thus, we can anticipate with limited simulation budget, end-to-end DSI nosie mechanism on SGD cannot provide meaningful utility-trustworthiness tradeoff.

Therefore, in the proposed DSI deep learning framework, we still adopt an iterative perturbation method through composition, with efficiency consideration.

### E.2. Isotropic DP Noise and Anisotropic DSI Noise Bounds

Assisted with sampling amplification, under a $T$-iteration budget where each datapoint in an $n$-element dataset is selected with probability $q$ in each batch, when the sampling rate $q$ is sufficiently large, specifically $\omega(\sqrt{\epsilon/T})$, DP-SGD requires per-iteration noise with an expected $l_2$-norm of $O\left(\frac{\sqrt{dT \log(1/\delta)}}{n}\right)$ to ensure a global $(\epsilon, \delta)$ DP guarantee (Abadi et al., 2016). In practical scenarios, DP-SGD typically achieves best performance when a large, constant subsampling rate $q$ is used (De et al., 2022), as smaller subsampling rates may fail to fully exploit linear amplification effects (Zhu & Wang, 2019).

In the presented DSI analysis, we do not account for amplification arising from the randomness of subsampling in the iterative deep learning algorithm. Strictly speaking, this makes our noise bound conservative. However, this simplification also grants us the flexibility to freely design our mini-batch selection strategy. For instance, consider a standard partitioning approach where the dataset of $n$ samples is divided into $1/q$ batches, each containing $nq$ samples. In this setup, each datapoint appears in $Tq$ of the total $T$ iterations, requiring only a $Tq$-composition analysis.

For each iteration, the $l_2$-norm of the required per-iteration noise is scaled at most by $\sqrt{nq}$, even in the worst-case scenario where the differences among the $nq$ leave-one-out subsets (corresponding to the $nq$ selected samples) are orthogonal to each other. Combining these considerations, the expected $l_2$-norm of the per-iteration noise required for a global $(\epsilon, \delta)$ guarantee under the DSI noise framework, with a $Tq$ composition, is given by:

$$O\left(\frac{1}{nq} \cdot \sqrt{nq} \cdot \frac{\sqrt{Tq \log(1/\delta)}}{\epsilon}\right) = O\left(\frac{\sqrt{T \log(1/\delta)}}{\sqrt{n}\epsilon}\right). \tag{35}$$

It is worth noting that the bound in (35) is independent of the batch size.

## F. Experiments

### F.1. Comparison with DP-SGD

In Table 1, we implement DSI local SGD as follows: for each epoch, the CIFAR-10 training set of 50,000 samples, as our input set $U$, is partitioned into 125 batches, each containing 400 samples. In the $t$-th round, we further divide each batch into $L = 20$ subgroups, $G_1^{(t)}, G_2^{(t)}, \ldots, G_{20}^{(t)}$, each consisting of 20 samples. We then perform $K = 20$ local full gradient descent steps using optimizer $\mathcal{O}$, initialized from the preceding iterate $w^{(t-1)}$, on each subgroup $G_l^{(t)}$. The final output of this round is obtained by averaging the updated weights from all $L = 20$ subgroups as:

---

[4] We apply a batch size of 128 and run SGD for 200 epochs.

| Duplication($v$), model | unseen data | $\epsilon = \infty$ | $\epsilon = 2$ | $\epsilon = 4$ | $\epsilon = 6$ |
|---|---|---|---|---|---|
| 1, GPT2-small | 1.66 | 3.15 | 2.05 | 2.25 | 2.55 |
| 10, GPT2-small | 1.66 | 9.80 | 2.60 | 4.20 | 5.60 |
| 1, OPT-125M | 2.05 | 2.90 | 1.70 | 1.55 | 2.35 |
| 10, OPT-125M | 2.05 | 17.9 | 3.20 | 7.60 | 8.50 |
| 1, OPT-350M | 1.23 | 3.85 | 2.15 | 2.25 | 2.65 |
| 10, OPT-350M | 1.23 | 39.2 | 3.90 | 4.50 | 9.50 |

*Table 4.* $(v, 100, 5)$ **Exact Memorization Rate (%)** when finetuning different LLMs using WikiText5 with and without ($\epsilon = \infty$) DSI guarantees in $(\epsilon, \delta = 10^{-5})$.

$$w^{(t)} = \frac{1}{20} \sum_{l=1}^{20} \mathcal{O}(w^{(t-1)}, G_l^{(t)}). \tag{36}$$

For a reference subset $\mathsf{R}_i$, note that in each round, the computation over $\mathsf{R}_i$ differs from that over $\mathsf{U}$ in at most one local gradient descent step—specifically, in the subgroup containing the differing datapoint $\mathsf{U} \setminus \mathsf{R}_i$. Thus, to compute the noise in each iteration, it suffices to determine $z_j$ for $j = 1, 2, \ldots, 200$, where $z_j$ corresponds to the outcome difference:

$$z_j = \frac{1}{20} \big( \mathcal{O}(w^{(t-1)}, G_i^{(t)}) - \mathcal{O}(w^{(t-1)}, G_i^{(t)} \setminus u_j) \big), \tag{37}$$

for each $u_j$ included in subgroup $G_i^{(t)}$.

This subgrouping strategy is primarily motivated by implementation efficiency. Under this setup, determining the noise per round requires only 201 executions of 20 local gradient descent steps over 10 samples, compared to 201 executions over 200 samples, significantly reducing computational overhead.

As for the hyperparameter selection, for $\epsilon = 1$, $\epsilon = 2$, $\epsilon \in [3:4]$, and $\epsilon \in [5:8]$, we select the epoch number to be 3, 6, 10 and 15, respectively. We uniformly select the learning rate of the local SGD to be $1.25 \cdot 10^{-4}$.

### F.2. Memorization

We construct the WikiText-5 dataset by taking the first 5/103 portion of text examples from WikiText-103 (Merity et al., 2016), resulting in a training set of approximately 5.7 million tokens. The samples are then reformatted into 5,640 sequences, each of length 1,024, with each sequence treated as an individual sample.

For different models, we determine the optimal number of training epochs based on validation perplexity, selecting the highest-performing epoch before overfitting occurs given the chosen learning rate. Specifically:

- For GPT-2 small, we use a learning rate of $3 \times 10^{-4}$ and train for 1 epoch.

- For OPT-125M, we use a learning rate of $1.5 \times 10^{-4}$ and train for 1 epoch.

- For OPT-350M, we use a learning rate of $1 \times 10^{-5}$ and train for 5 epochs.

For all DSI-related results presented in Fig. 2 and Tables 2 and 4, we employ standard Adam (Kingma, 2014), with a batch size of 20 *without* local iterations, as our optimizer $\mathcal{O}$ within the proposed DSI deep learning framework.

### F.3. Backdoor Attacks

We consider the following setup: for $m$ sources/entities where one of them is malicious, we evenly split the CIFAR10 training dataset, totally 50,000 samples, into $m$ disjoint subsets, each with $50000/m$ samples; for the objective poisoning strategy, the malicious entity will transform their assigned samples accordingly, while all other $(m-1)$ users hold the original clean samples assigned. The PreAct-ResNet18 is pre-trained using CIFAR100 data.

The federate learning algorithm $\mathcal{F}$ is defined as follows. The entire training procedure consists of $T$ phases, and each phase, captured by $\mathcal{O}$ in Algorithm 2, is formed by $K$ local iterations. In each iteration, the central server asks each participated entity to subsample $2000/m$ many samples from their local data and accordingly compute and send the gradient computed on current iterate. The server then *robustly* aggregate the received gradients in three steps:

1. **Clipping**: Suppose the server receives $n$ gradients $g_1, g_2, \cdots, g_n$ from each participated entity. They first determine

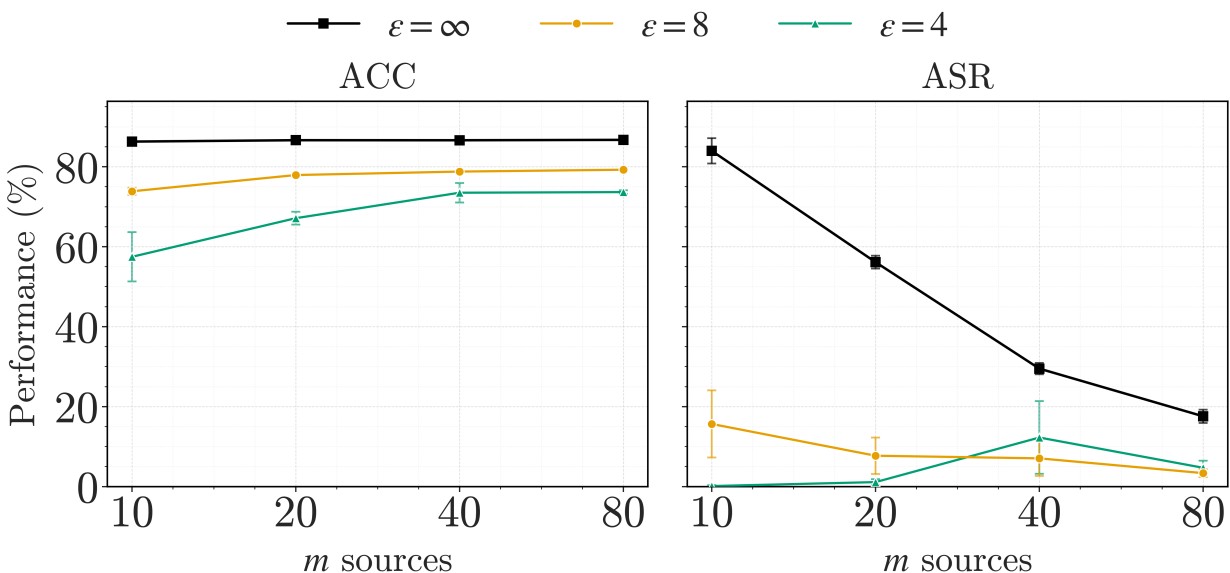

*Figure 4.* **Test Accuracy** (ACC) and **Attack Success Rate**(ASR) when training poisoned CIFAR10 under **Blended** attacks (Chen et al., 2017) with/without DSI guarantees ($\epsilon, \delta = 10^{-5}$).

the medium of the $l_2$-norm of the $n$ entity-level gradient, denoted by $\bar{s}$. Then, each gradient $g_i$ is clipped to $g_i' \leftarrow g_i \cdot \min\{1, \frac{\|g_i\|}{\bar{s}}\}$.

2. **Cosine-Similarity Projection**: For clipped gradients $g'_{[1:n]}$, the server computes their mean, denoted by $\bar{g} = 1/n \cdot \sum_{i=1}^n g_i'$ and determine the angle $\theta_i$ between $g_i$ and $\bar{g}$, where $\cos(\theta_i) = \frac{\langle g_i', \bar{g} \rangle}{\|g_i\|\|\bar{g}\|}$. Similarly, we compute the medium of $\theta_{[1:n]}$, denoted by $\bar{\theta}$. We then further project each clipped $g'_{[1:n]}$ to a cone such that its projected version will not be more than $\bar{\theta}$ cosine-away from $\bar{g}$. To be formal,

$$g_i'' = \arg\min_g \|g - g_i'\|, \ s.t. \ \|g\| = \|g_i'\|, \arccos(\frac{\langle g_i', g \rangle}{\|g_i\|\|\bar{g}\|}) \geq \bar{\theta}. \tag{38}$$

.

3. **Aggregation and Update**: Finally, the server takes the average of $g''_{[1:m]}$ as the update and move forwards to the next iteration.

The above-described operations ensure that the gradient collected from each entity cannot be arbitrarily large or pointing out to some direction dramatically different from the majority. This simple robust aggregation is helpful in applying DSI noise to avoid the trivial attack where the adversary may just send large outliers to enforce adding a huge noise to destroy the utility.

In particular, for the hyper-parameter selection in our experiments shown in Fig. 3 and 4 we select $T = 50$ and $K = 10$ with a step size (learning rate) $10^{-3}$. As mentioned before, each reference set $\mathsf{R}_i$ is selected to be the entire training data $\mathsf{U}$ excluding the samples from $i$-th entity. Thus, for each phase, to determine the outcome difference $z_i$, we will implement $\mathcal{O}$ across $\mathsf{U}$ and $\mathsf{R}_{[1:m]}$, with a computation overhead $O(m^2)$. [5]

### F.4. Comparison between DSI Noise and Isotropic Noise in Defending Backdoor Attacks

In this subsection, we further include the comparison of the *empirical* efficiency between classic isotropic noise and the optimized anisotropic noise in defending backdoor attacks.

**Matching Utility Performance**: We first adjust the variance of isotropic noise to match the test accuracy obtained with the DSI framework under $\epsilon = 4$ and $\epsilon = 8$ with $\delta$ fixed to be $10^{-5}$. We then compare adversarial success rate (ASR), showing

---

[5]When we implement $\mathcal{O}$ on each $\mathsf{R}_i$, each local iteration only takes the gradient collected from $(m-1)$ entities; while when we implement $\mathcal{O}$ on $\mathsf{U}$, each local iteration takes the gradient collected from all $m$ entities.

that for the same empirical utility, DSI noise on average has a better efficiency compared to isotropic noise in defending against backdoor attacks, as shown in Table 5, 6.

**Evaluating Provable Indistinguishability**: We also report the provable $(\epsilon, \delta)$ guarantees achievable by isotropic noise in this setup. As expected, when the number of sources $m$ is small, worst-case sensitivity remains high. Even with $m = 80$, for blended attacks, the resulting bound ($\epsilon = 179, \delta = 10^{-5}$) is too weak to provide meaningful guarantees, to match the same performance from DSI framework with ($\epsilon = 4, \delta = 10^{-5}$).

*Table 5.* Comparison on indistinguishability control and defense efficiency in Adversarial Success Rate (ASR) against **Low-Frequency Attacks** (Zeng et al., 2021) between **DSI Noise** and **Isotropic Noise** (Iso.) with **fixed** test accuracy on clean data.

| (a). $m = 10$ Sources | | |
|---|---|---|
| Test ACC (%) | Ind. Guarantee | ASR (%) |
| 75.6 | ($\epsilon = 8, \delta = 10^{-5}$) (DSI) | **13.9** |
| | ($\epsilon = 2019, \delta = 10^{-5}$) (Iso.) | 25.6 |
| 65.5 | ($\epsilon = 4, \delta = 10^{-5}$) (DSI) | **2.0** |
| | ($\epsilon = 575, \delta = 10^{-5}$) (Iso.) | 20.2 |

| (b). $m = 20$ Sources | | |
|---|---|---|
| Test ACC (%) | Ind. Guarantee | ASR (%) |
| 79.1 | ($\epsilon = 8, \delta = 10^{-5}$) (DSI) | **8.4** |
| | ($\epsilon = 1661, \delta = 10^{-5}$) (Iso.) | 16.2 |
| 70.9 | ($\epsilon = 4, \delta = 10^{-5}$) (DSI) | **5.3** |
| | ($\epsilon = 479, \delta = 10^{-5}$) (Iso.) | 11.5 |

| (c). $m = 40$ Sources | | |
|---|---|---|
| Test ACC (%) | Ind. Guarantee | ASR (%) |
| 78.4 | ($\epsilon = 8, \delta = 10^{-5}$) (DSI) | 6.4 |
| | ($\epsilon = 627, \delta = 10^{-5}$) (Iso.) | **7.3** |
| 73.6 | ($\epsilon = 4, \delta = 10^{-5}$) (DSI) | **6.6** |
| | ($\epsilon = 312, \delta = 10^{-5}$) (Iso.) | 7.2 |

| (d). $m = 80$ Sources | | |
|---|---|---|
| Test ACC (%) | Ind. Guarantee | ASR (%) |
| 79.5 | ($\epsilon = 8, \delta = 10^{-5}$) (DSI) | 8.9 |
| | ($\epsilon = 350, \delta = 10^{-5}$) (Iso.) | **8.2** |
| 74.1 | ($\epsilon = 4, \delta = 10^{-5}$) (DSI) | **6.1** |
| | ($\epsilon = 114, \delta = 10^{-5}$) (Iso.) | 7.7 |

*Table 6.* Comparison on indistinguishability control and defense efficiency in Adversarial Success Rate (ASR) against **Blended Attacks** (Chen et al., 2017) between **DSI Noise** and **Isotropic Noise** with **fixed** test accuracy on clean data.

| (a). $m = 10$ Sources | | |
|---|---|---|
| Test ACC (%) | Ind. Guarantee | ASR (%) |
| 73.9 | ($\epsilon = 8, \delta = 10^{-5}$) (DSI) | 15.7 |
| | ($\epsilon = 2019, \delta = 10^{-5}$) (Iso.) | **9.9** |
| 57.5 | ($\epsilon = 4, \delta = 10^{-5}$) (DSI) | **0.1** |
| | ($\epsilon = 350, \delta = 10^{-5}$) (Iso.) | 9.8 |

| (b). $m = 20$ Sources | | |
|---|---|---|
| Test ACC (%) | Ind. Guarantee | ASR (%) |
| 77.9 | ($\epsilon = 8, \delta = 10^{-5}$) (DSI) | **7.7** |
| | ($\epsilon = 1661, \delta = 10^{-5}$) (Iso.) | 8.8 |
| 67.1 | ($\epsilon = 4, \delta = 10^{-5}$) (DSI) | **1.1** |
| | ($\epsilon = 241, \delta = 10^{-5}$) (Iso.) | 2.4 |

| (c). $m = 40$ Sources | | |
|---|---|---|
| Test ACC (%) | Ind. Guarantee | ASR (%) |
| 78.8 | ($\epsilon = 8, \delta = 10^{-5}$) (DSI) | 7.1 |
| | ($\epsilon = 627, \delta = 10^{-5}$) (Iso.) | **4.1** |
| 73.5 | ($\epsilon = 4, \delta = 10^{-5}$) (DSI) | 12.3 |
| | ($\epsilon = 312, \delta = 10^{-5}$) (Iso.) | **5.1** |

| (d). $m = 80$ Sources | | |
|---|---|---|
| Test ACC (%) | Ind. Guarantee | ASR (%) |
| 79.3 | ($\epsilon = 8, \delta = 10^{-5}$) (DSI) | **3.4** |
| | ($\epsilon = 350, \delta = 10^{-5}$) (Iso.) | 3.9 |
| 73.7 | ($\epsilon = 4, \delta = 10^{-5}$) (DSI) | 4.7 |
| | ($\epsilon = 179, \delta = 10^{-5}$) (Iso.) | **3.6** |

