# OpenReview forum: "Trustworthy Machine Learning through Data-Specific Indistinguishability"
_ICML.cc/2025/Conference — ICML 2025 poster_

### Official Review · Reviewer_9rXN · 2025-03-12

**Overall Recommendation:** 3

**Summary:**

This paper proposes a concept of (gaussian) data-specific indistinguishability (DSI), which relaxes Input-Independent Indistinguishability (or differential privacy in many sense) by enforcing constraints only for a set of pre-defined input pairs instead of globally.

Similar to what we already have for differential privacy, they derive the corresponding Gaussian mechanism, composition rules and group versions for DSI, which consequently allows using Gaussian noise in parameter optimizations (SGD) to ensure DSI of the entire scheme (just like DP-SGD for differential privacy in many ways). They denote this as DSI deep learning.

Experimentally, they show example application of DSI deep learning in reducing memorization of fine-tuned language models and defending backdoor attacks in federated learning.

---

### update after rebuttal
While DSI is an interesting notion different from existing ones, I am not yet convinced if/how DSI is more preferable in applications suggested by the authors. I can see how DSI can lead to better provable (robustness) bounds than differential privacy, but I think the bounds are still not meaningful practically in most cases as $e^\epsilon$ can be large even for single-digit $\epsilon$ and there are established methods with better provable robustness (e.g. DPA or bagging or Finite Aggregation for provable backdoor/poisoning defenses).

I am ok with it being accepted but will not champion for it based on current materials provided, which still aligns with my initial recommendation of "3: Weak accept (i.e., leaning towards accept, but could also be rejected)".

**Claims And Evidence:**

Most claims are well supported, except:
1. In section 4 and in conclusion, it is claimed that "DSI local-SGD outperforms DP-SGD in all cases", "our initial results demonstrate a significant improvement in the utility-trust trade-off compared to traditional, such as DP-based, methods", which is not supported by existing evidence. While in table 1 it is showed that DSI-Local-SGD can offer higher accuracy than DP-SGD **assuming the same $\epsilon$ and $\delta$**, using the same $\epsilon$ and $\delta$ for differential privacy and DSI does not indicate the same level of trust/privacy/indistinguishability, rendering the arguments unsupported. Naturally, this issue also affect many other claims made in section 4 involving advantage of DSI (DSI-local SGD) over DP (DP-SGD).
2. Section 5 experiments: "for all the following experiments with (ε, δ) parameters considered DP-SGD always requires prohibitively large noise which fully destroys the performance." This is not well justified, due to the very same issue as the one above.

**Essential References Not Discussed:**

Not that I am aware of.

**Experimental Designs Or Analyses:**

The experimental designs for (reducing) memorization in LLM and (defending) backdoor attacks in federated learning are ok, with a flaw that no baseline is compared to, including but not limited to DP-SGD.

The primary issue in existing experiments across different parts is that the submission assumes, without good justifications, that different privacy and the proposed DPI should be compared by simply assuming the same epsilon and delta.

**Methods And Evaluation Criteria:**

Yes.

**Other Comments Or Suggestions:**

minor:
The running title of the submission is not updated.

**Other Strengths And Weaknesses:**

[strength]
The idea of relaxing the global requirement for Input-Independent Indistinguishability/differential privacy is very natural, and the authors put in great efforts in providing a fairly comprehensive set of theoretical tools/results, which is impressive.

[weakness]
The primary weakness is the limited investigation (regardless of theoretical/empirical) regarding the degree of trust/privacy/robustness provided by the proposed DSI notions, especially in comparison with differential privacy. Basically there is no clear indications regarding whether/why DSI is a more useful notion compared to differential privacy, for example.

**Questions For Authors:**

The primary question/concern I have is simply: What supports are there indicating DSI is promising/a better tool than e.g. differential privacy?

As there is no results aligning/comparing the effectiveness (i.e. the degree of trust/privacy/protection they provide) of DSI & differential privacy, this remains unclear and renders the assessment quite tricky.

**Relation To Broader Scientific Literature:**

I can see the proposed DSI framework is very closely related to differential privacy, with many tools/results most likely adapted or at least inspired from existing, well-established results for differentially private deep learning research.

Notably, the suggested applications of DSI, i.e. reducing memorization and mitigating backdoor attacks, also overlap with applications previous suggested for differential privacy.

**Theoretical Claims:**

While I did not check the proofs line by line, I skimmed through the main theoretical results and they made sense.

---

> ### Author Rebuttal · Authors · 2025-04-01
>
> Thank you for your positive assessment and valuable suggestions.
>
> **1. Differential Trustworthiness (DT) vs. Differential Privacy (DP)**
>
> We fully agree with your insightful comment: “Data-Specific Indistinguishability (DSI) and Differential Privacy (DP) (or Input-Independent Indistinguishability (III)) **even under the same** parameters $(\epsilon, \delta)$ do **not** imply the same level of trust or privacy.” In Section 1.2 and our Response 1 to Reviewer 53pn, we explain why III is necessary for privacy—instance-based privacy statements or noise mechanisms themselves may already leak information.
>
> The goal of this paper is **not** to relax existing privacy frameworks (e.g., DP) or improve privacy solutions. Instead, we focus on a broad class of trust concepts that are **not** primarily concerned with preventing information leakage but rather with **controlling data usage** (e.g., copyright protection), **mitigating influence** (e.g., backdoor defense), and **governing model behavior** (e.g., memorization mitigation).
>
> A key insight from our Differential Trustworthiness (DT) framework is that if we can bound the distinguishability between a target model and a set of safe reference models—such as a model trained on clean data (for backdoor defense) or a dataset excluding memorization-sensitive information—we can obtain a probabilistic guarantee that the target model meets objective trustworthiness requirements.
>
> More importantly, when information leakage is not a concern, input independence is **not** necessary to derive DT guarantees. This explains why DP (which enforces III) is a **sufficient but not necessary** condition for DT. Our framework establishes an optimal noise mechanism to achieve DSI, significantly more efficient than **existing distinguishability control** methods in DP. But, in Section 1.2 including Footnote 2, we clarify DSI noise mechanisms **cannot** be used to address privacy concerns or provide DP-like guarantees, and we will emphasize this further in our revision.
>
> **2. Efficiency of Indistinguishability Control**
>
> The goal of our experiments is indeed to compare the **efficiency of achieving indistinguishability** using DP's standard methods—**clipping (sensitivity control) and isotropic noise**—versus our DSI framework for black-box data processing. By targeting the same divergence bound (measured in Hockey-Stick divergence with $(\epsilon, \delta)$ parameters), we demonstrate the significant utility improvement enabled by our optimized noise. We will clarify and emphasize that we are **not** comparing the privacy guarantees ensured by DP-SGD in the revision.
>
> **3. Baseline and Additional Experiments Comparing Isotropic and DSI Noise**
>
> We apologize for any misunderstanding regarding our notation. In all experiments, we include a baseline (see the column with $\epsilon = \infty$), which represents the original case **without** any perturbation.
>
> Additionally, we provide **new experiments in Exp 3 in the attachment (https://anonymous.4open.science/r/4722-1FD5)**. Specifically, in a backdoor defense scenario (see Appendix F.3) where training data is collected from $m$ sources (one of which is malicious), we compare the trust-utility tradeoff achieved via DP-SGD (gradient clipping + isotropic noise) versus the DSI framework.
>
> - 3-1) Matching Utility Performance: For $m$ varying from $10$ to $80$, we first adjust the variance of isotropic noise to match the test accuracy obtained with the DSI framework under $(\epsilon=4, \delta=10^{-5})$ and $(\epsilon=8, \delta=10^{-5})$, respectively.  We then compare the adversarial success rate (ASR), showing to produce the **same empirical test accuracy**, DSI outperforms isotropic noise (leading to lower ASR rates) in defending against backdoor attacks in most cases.
>
> - 3-2) Evaluating Provable Indistinguishability: We also report the provable $(\epsilon, \delta)$ guarantees achievable by isotropic noise in this setup. As expected, given that the number of data sources $m$ is relatively small, the worst-case sensitivity $O(1/m)$ remains high. Even with $m=80$, the resulting bounds **$(\epsilon=114, \delta = 10^{-5})$** for low-frequency attacks [1] and **$(\epsilon=179, \delta = 10^{-5})$** for Blended attacks [2] are too weak to provide meaningful guarantees, to match the same performance from DSI framework with  $(\epsilon=4, \delta=10^{-5})$.
>
> These results will be included in our revision, further supporting our claim that standard mechanisms in DP-SGD cannot produce usable divergence bounds in our experimental settings (especially for small dataset and scaling/high-dimensional models).
>
> Finally, we would greatly appreciate your feedback on whether we have adequately addressed your concerns. Please let us know if you have any additional questions.
>
> [1]. Zeng, Yi, et al. "Rethinking the backdoor attacks' triggers: A frequency perspective."
>
> [2]. Chen, Xinyun, et al. "Targeted backdoor attacks on deep learning systems using data poisoning."

---

> > ### Comment · Reviewer_9rXN · 2025-04-04
> >
> > My primary concerns in the initial review can be summarized as:
> > What supports are there indicating DSI is promising/a better tool than e.g. differential privacy? There is no clear indications regarding whether/why DSI is a more useful notion compared to differential privacy, for example.
> >
> > The point 1 & 2 in the rebuttal are simply the differences between DSI and differential privacy rather than DSI's advantages, which means point 1 & 2 in the rebuttal are not really helpful in addressing these concerns.
> >
> > The results in new experiments in Exp 3 in the attachment (point 3) are very related to the concerns, specifically comparing empirically DSI and DP-SGD as defenses. However, I find it difficult to agree with the authors' interpretation of the results:
> >
> > >From rebuttal: "We then compare the adversarial success rate (ASR), showing to produce the same empirical test accuracy, DSI outperforms isotropic noise (leading to lower ASR rates) in defending against backdoor attacks in most cases."
> >
> > In fact, according to exp 3, DP-SGD performs better than the proposed DSI framework in **2 out of 8** cases in Table 4 and **4 out of 8** cases in Table 5. These are for sure not conclusive/significant enough to claim the advantage of DSI notions over differential privacy.
> >
> > To sum up, while DSI is an interesting notion different from existing ones, I am not yet convinced that DSI is more preferred in applications suggested by the authors. I am ok with it being accepted but will not champion for it based on current materials provided, which still aligns with my initial recommendation of "3: Weak accept (i.e., leaning towards accept, but could also be rejected)".

---

> > > ### Author Response · Authors · 2025-04-05
> > >
> > > Thank you very much for your response and for sharing your concerns!
> > > We’d like to clarify some of the points and potential confusions in the following.
> > >
> > > 1. **Sharpened Utility-Trust Tradeoff with Data-Specific Indistinguishability (DSI)**
> > >
> > > The **key advantage**/contribution of DSI lies in its ability to provide **provable** trust guarantees while incurring significantly **lower utility overhead** compared to Differential Privacy (DP) or Input-Independent Indistinguishability.
> > >
> > > To illustrate this, consider the memorization mitigation of a particular data point $x_0$​. Both DP and DSI (reference set defined as leave-one subset, i.e., removing $x_0$​)—when having the same indistinguishability parameters $(\epsilon_0, \delta_0)$—actually lead to the **same** guarantee: if the probability that the reference model (trained without $x_0$​) memorizes $x_0$ is $p_0$​, then the probability that the target model (trained with $x_0$​) memorizes $x_0$​ is upper-bounded by $$e^{\epsilon_0}p_0 + \delta_0.$$ Please refer to our memorization experiments for more details (Section 5a-1: memorizing 6-digit strings; Section 5a-2: memorizing token subsequences).
> > >
> > > We also verify the tightness of the probabilistic bounds based on distinguishability (see left column, lines 405–408 and 432–435). Thanks to optimized noise and tighter accounting methods  (composition and grouping), our theoretical bounds via DSI—**though still conservative**—substantially sharpen the tradeoff between trust (indistinguishability) and utility. For example, in LLM fine-tuning experiments on Wikitext-5 (a relatively small dataset), achieving $(\epsilon=8, \delta=10^{-5})$ with DP-SGD requires prohibitively-large noise due to high model dimensionality (hundreds of millions of parameters), resulting in unacceptable performance (perplexity > 40). In contrast, DSI achieves meaningful guarantees with much better utility (perplexity between 16–22; see Table 2).
> > >
> > > Our additional experiments on backdoor defense further highlight DSI's efficiency for **provable distinguishability control**. As shown in Tables 3 and 4 in the attachment:
> > > To match the utility of DSI, the $(\epsilon, \delta)$ values that can be ensured by DP-SGD can be **hundreds of times larger**, which are too weak to produce any meaningful guarantees. Conversely, to enforce a reasonable single-digit $\epsilon$ using DP-SGD, test accuracy on CIFAR-10 drops to ~20%, compared to ~70% using DSI.
> > >
> > > 2. **Additional Operational Advantages of DSI**
> > >
> > > Beyond efficiency, DSI offers several practical advantages over DP:
> > > - a) **Black-box Processing and Customized Budgets**: DSI can be applied in black-box settings, and supports differentiated distinguishability budgets across multiple references. In contrast, DP requires white-box sensitivity analysis and can only capture the worst case. The flexibility of DSI allows, for example, assigning different $(\epsilon,\delta)$ budgets to control contribution from mutiple data sources—useful in scenarios like copyright or contribution attribution, which DP cannot directly support.
> > >
> > >
> > > - b) **Modeling More General Differential Trust**:  DSI is not necessarily restricted to leave-one-out reference sets, as considered in DP.  The reference sets in DSI can be very general. We provide a possible use case: Imagine multiple companies train separate LLMs using their own data and algorithms. For privacy consideration, they all want to ensure the generated responses on an arbitrary query from their models do **not** reveal any **unique** information containing in their training data. A DSI solution can be adding an optimized noise to (semantic embedding of) each response on a given query, which mitigates their difference while maximally preserving their common patterns or information.
> > >
> > > - c) **One-Way Divergence**: As mentioned in Section 3.2(i), DSI only requires one-way divergence control, while DP necessitates two-way worst-case analysis, which is more complex and less flexible [1].
> > >
> > > 3. **On Interpretation of New Experimental Results**
> > >
> > > We apologize for any unintended implications for the added experiments on backdoor defense. We faithfully report the empirical defense performance of both DSI and DP noises as a complement. However, as noted, our main focus is always on the efficiency to achieve **provable** trust/indistinguishability and we do **not** plan to claim any empirical superiority of DSI noise itself.
> > >
> > >
> > > Finally, thank you once again for your thoughtful feedback. To the best of our knowledge, DSI is the first attempt to **systematically** unify a wide range of differential trust concepts and efficiently build **provable** guarantees. The combination of optimized DSI noise and improved accounting methods meaningfully bridges the **gap between theory and practice** in cutting-edge trustworthy AI research. We would deeply appreciate your support on this new research direction.
> > >
> > > [1] Zhu, Yuqing, and Yu-Xiang Wang. "Poission subsampled rényi differential privacy."

---

### Official Review · Reviewer_cmFq · 2025-03-13

**Overall Recommendation:** 2

**Summary:**

The paper aims at combining various privacy preserving mechanism for machine learning into a unified framework, for instance reducing memorization, providing copyright protection, differential privacy and so on. The main mathematical technique is to use Data Specific Indistinguishability which aims at providing privacy protection to be dependent on the data (training data) and its restrictions instead of the worst case bounds in DP. The propose an algorithm to achieve this and discuss its properties like composition, post processing etc.

**Claims And Evidence:**

The main claim is providing a DSI procedure which can encompass various privacy preserving mechanisms into one definition and making it data dependent more specifically.

Section 3 discusses the DSI gaussian mechanism, which aims to add gaussian noise as a way of providing privacy, and providing data dependent results i.e. removing the data independence assumption used in DP

In Section 3.2 the paper discusses various properties, and shows how the proposed method satisfies them.

In Algorithm 2, the paper proposes a framework for optimizing deep learning model with black box optimizers by modifying the gradients in similar spirit to DP-SGD

**Essential References Not Discussed:**

Some essential references which are missing:

Per-instance DP:
- https://arxiv.org/abs/1707.07708
- https://arxiv.org/abs/2111.02281
- https://openreview.net/pdf?id=ESt7ECoWpn

Subspace based DP:
- https://arxiv.org/abs/2108.11527
- https://www.math.uci.edu/~rvershyn/papers/hsvz-subspaceprivacy.pdf

Public-Private DP:
- https://arxiv.org/pdf/2306.15056
- https://arxiv.org/pdf/2203.11481
- https://proceedings.mlr.press/v202/nasr23a/nasr23a.pdf

PAC Privacy:
- https://arxiv.org/abs/2210.03458
- https://arxiv.org/abs/2312.01201

Data dependent DP
- https://arxiv.org/pdf/1905.12813
- https://papers.nips.cc/paper_files/paper/2018/hash/9a0ee0a9e7a42d2d69b8f86b3a0756b1-Abstract.html

**Experimental Designs Or Analyses:**

I think the baselines used are not up to the mark, and its not the best to compare against DP-SGD. There are works on subspace identification in DP, where we can minimize the damage caused by clipping and gaussian noise.

The major concern I have is that the related works have not been correctly used to build upon. There have been plenty of works for improving DP.

Here are few examples:
https://arxiv.org/pdf/2007.03813
https://cdn.aaai.org/ojs/20315/20315-13-24328-1-2-20220628.pdf
https://arxiv.org/abs/1707.07708
https://arxiv.org/pdf/2203.11481

**Methods And Evaluation Criteria:**

The proposed method is relevant and interesting, however, for the experiments / baselines, it appears that the paper does not compare against the right baselines. For instance, there have been several papers on data dependent DP, which exploit subspaces to improve the utility of the underlying privacy mechanism. However, the only experiment / baseline was DP-SGD

Theoretically, it seems to be a relaxation of DP-SGD where the gradient clipping is by-passed as part of the definition, and the addition of noise is data dependent.

In the conclusion the authors mention that their approach is significantly better than DP, however, they both are aimed at doing different things. And to make a fair comparison it necessary to compare to the right papers.

Some relevant baselines:
https://arxiv.org/pdf/2311.14632
https://arxiv.org/pdf/2210.00036
https://proceedings.mlr.press/v202/bu23a/bu23a.pdf
https://arxiv.org/abs/2212.00328
https://arxiv.org/pdf/2203.11481

**Other Comments Or Suggestions:**

I will recommend the authors to spin the paper as an unlearning / obfuscation paper rather than differential trustworthiness because privacy in general is too broad, and there is a lot of work in relaxing DP which makes the novelty of the proposed approach low.

Also, there is no limitations section in the paper. One trivial limitation I can see is that the proposed method has no worse case bounds which can be of utmost importance in privacy. I will encourage the authors to add a section the limitations of the proposed method.

**Other Strengths And Weaknesses:**

Strengths:
- The paper is well written, and the theory is easy to parse and grounded.
- The problem is very interesting to work on, as it corresponds to reducing multiple sub-problems in privacy preserving machine learning to a single optimization problem
- Using data dependent anisotropic gaussian noise for privacy preservation is intuitive, and in general having data dependent privacy is the way for the future.
- The proposed approach seems to be more relevant for unlearning, obfuscation than privacy in general whose definition can vary a lot depending on the application.

Weaknesses:
- The proposed methods are similar to the existing methods in DP literature, so the novelty of the proposed appears to be low.
- The baselines considered are weak, and thus it would be useful if it can be improved.

**Questions For Authors:**

I think if the authors provide a detailed comparison against methods like per-instance DP, DP in public-private setting, PAC-privacy that will be great. The current method seems to be very correlated with existing methods, and thus its essential to make it clear what is the major contribution which is different from the past works.

**Relation To Broader Scientific Literature:**

The paper aims provide a framework for differential trustworthiness, and provide a data dependent definition for it. It is well know in the privacy community that DP implies reduced memorization, copyright protection (using NAF), unlearning, and robustness to membership attacks. However, its difficult to make DP work in practice at scale, as the amount of noise scales rapidly with size, and thus the models trained with DP have reduced utility.

The paper aims to relax this and provide a simple procedure where we add gaussian noise which is tailored to the training data. While the aim and direction of the paper is interesting, I think there are a lot of closely related works which do something similar.

Also, the aim is at providing differential trustworthiness, however, its eventually compared against DP-SGD

**Theoretical Claims:**

The definitions and theorems seem to be sound.

---

> ### Author Rebuttal · Authors · 2025-04-01
>
> Thank you for your positive assessment and helpful suggestions.
>
> **1. Relationship Between Privacy and Differential Trustworthiness (DT)**
>
> We fully agree with your insightful comment that (differential) privacy is recognized as a stronger guarantee, which, as a **sufficient** condition, can produce many other (differential) trust guarantees (e.g., reducing memorization and protecting copyright). However, privacy is costly (we also acknowledge your point that "privacy" is a broad concept; here, we specifically use "privacy" to refer to confidential protection). This motivates us to explore more efficient solutions to address “weaker” trust guarantees—those aimed at controlling the influence of training data on models without necessarily preventing data leakage.
>
> Strictly speaking, privacy is **not** directly comparable to the DT (memorization, backdoors, copyright). One key insight we highlight (see Sections 1.1 and 1.2) is that privacy based on **Input-Independent Indistinguishability (III)** form a sufficient **but not necessary** solution: when the goal is not to prevent information leakage, input independence is **unnecessary** and technically only indistinguishability guarantee is needed to build provable trust. Thus, we propose and justify a relaxed version, Data-Specific Indistinguishability (DSI), and construct the optimal DSI noise mechanism. Due to space constraints, please refer to our Response 1 to Reviewer 53pn for more details on why input independence is necessary for privacy, how DSI compares to per-sample/individual Differential Privacy (DP).
>
> **2. Comparison to Distinguishability Control Tools rather than DP Definition**
>
> Our key motivation and contribution are **not** to improve or relax existing DP frameworks but rather to develop better methods for constructing indistinguishability for DT guarantees. As demonstrated above, DSI is strictly weaker than III and is also incomparable to DP guarantees. What our experiments actually target is the **efficiency of different mechanisms in achieving indistinguishability**.
>
> As the baseline, the classic method to control distinguishability is through **clipping (sensitivity control) combined with isotropic noise**, as widely explored in DP literature. DP-SGD is a representative. To compare, we propose a new **anisotropic** noise mechanism to determine the minimal noise necessary for specific indistinguishability. Our framework also enables black-box algorithm analysis without requiring sensitivity control, thereby eliminating clipping bias. Furthermore, we establish accounting methods, such as composition (Theorem 3.4) and grouping (Lemma 3.5), to more tightly convert indistinguishability guarantees into probabilistic DT guarantees.
>
> In summary, our baseline comparison focuses on the efficiency of existing DP methods in achieving the required indistinguishability or statistical divergence bounds **rather than their privacy guarantees themselves**. To clarify, we will revise claims such as "comparison with DP-SGD" to "comparison with clipping + isotropic noise methods in DP-SGD."
>
> **3. Comparison to other Works**
>
> - PAC Privacy [1]:  PAC Privacy leverages secret entropy and models privacy risk by an adversary’s posterior success rate in recovering the secret. Different from examining the correlation between secrets and leakage from a privacy perspective, DT and DSI do **not** assume or rely on any input distribution. Instead, we optimize noise based on reference safe models.
>
> - Data-Dependent DP: Sticking to classical DP definitions, [2] utilizes public knowledge of graph model structures to optimize privacy budget allocation across multiple releases. However, the noise mechanism for each release remains the standard Laplace mechanism.
>
> - Sub-space and Public-private DP:  Assisted by public data, existing works have considered optimizing clipping strategies, subspace projection, or data augmentation. Still, all existing approaches apply standard **isotropic** noise. **In Exp2 ( https://anonymous.4open.science/r/4722-1FD5),  we include a more detailed comparison** and show **even without** assuming public data, DSI-SGD has outperformed all prior benchmarks training on CIFAR10 from scratch. In addition, public data tricks can also save DSI noise, e.g. mixing training data with public samples [3] can reduce the divergence between the target and reference models.
>
> **4. Limitations**
>
> We will expand the discussion on limitations. As you noted, when both provable privacy and DT guarantees are required simultaneously, the weaker DSI is not applicable and III remains the only known method.
>
> Finally, please do not hesitate to let us know if we have addressed your concerns or you have additional questions.
>
> [1] PAC Privacy: Automatic Privacy Measurement and Control of Data Processing
>
> [2] Data-Dependent Differentially Private Parameter Learning for Directed Graphical Model
>
> [3] DP-Mix: Mixup-based Data Augmentation for Differentially Private Learning

---

### Official Review · Reviewer_pwit · 2025-03-14

**Overall Recommendation:** 4

**Summary:**

This paper proposes a unified framework for Differential Trustworthiness (DT), which models trust-related concerns in machine learning algorithms, such as memorization, data poisoning, and copyright issues. The framework aims to regulate the divergence between the outputs of a model trained on a target dataset and those trained on reference datasets, ensuring trustworthy model behavior. Specifically, it introduces the concept of Data-Specific Indistinguishability (DSI) and its implementation through a Gaussian noise mechanism to mitigate differential effects and protect sensitive information. The paper also provides a thorough exploration of the theoretical properties, algorithmic solutions, and practical applications of DSI, particularly in areas like large language models and computer vision, addressing challenges such as privacy, model robustness, and utility trade-offs.

**Claims And Evidence:**

1.The writing of the abstract is somewhat difficult to understand and should be revised and expanded according to the logic of the introduction.

2.The proposed method shows significant theoretical contributions, but questions arise regarding its practical applications (e.g., defending against backdoor attacks). How does it differ from existing noise-based methods, such as those proposed in [1] and other works?

3.Experiments conducted on large language models are convincing. However, why are copyright-related experiments not included?

[1] NCL: Textual Backdoor Defense Using Noise-augmented Contrastive Learning, ICASSP 2023

**Essential References Not Discussed:**

Perhaps works related to memorization, data poisoning, and copyright issues concerning noise should be cited.

**Experimental Designs Or Analyses:**

See Claim and Evidence point 3.

**Methods And Evaluation Criteria:**

The DSI framework is novel, and its application to memorization, data poisoning, and copyright concerns is meaningful.

**Other Comments Or Suggestions:**

N/A

**Other Strengths And Weaknesses:**

N/A

**Questions For Authors:**

N/A

**Relation To Broader Scientific Literature:**

N/A

**Theoretical Claims:**

The theory appears to be sound.

---

> ### Author Rebuttal · Authors · 2025-04-01
>
> Thank you for your positive assessment and insightful questions.
>
> **1. Comparison to other Noise-based Solutions**
>
> We appreciate your references to prior works that apply noise for trustworthy machine learning. Existing noise-based solutions typically face two key challenges: **a) High utility overhead** and **b) Lack of provable guarantees**.
>
> As a representative of (a) and the primary baseline in our paper, Differential Privacy (DP) mechanisms can be used to ensure provable indistinguishability, which can further imply other trust guarantees such as memorization prevention and unlearning. However, ensuring a meaningful indistinguishability bound (i.e., small security parameters $\epsilon, \delta$) usually requires adding prohibitively-large isotropic DP noise, which is well-known to impose a significant utility cost.
>
> On the other hand, a long line of empirical work explores noise-based defenses **without** formal guarantees. For example, [1] proposes adding noise to training data as a general defense against unknown textual backdoor attacks; [2] introduces random smoothing, which augments training samples with noisy versions to improve adversarial robustness of trained models; [3] adds small noise to gradients during training to resist membership inference attacks. While these approaches demonstrate strong empirical success, they lack theoretical bounds on how much noise is needed to provably prevent attacks or ensure trust guarantees.
>
> Our Data-Specific Indistinguishability (DSI) noise framework overcomes these limitations by determining the **minimal necessary** noise to ensure **provable** trustworthy guarantees while maintaining low utility overhead. The key idea can be summarized below.
>
> - First, given an algorithm $\mathcal{F}$, we identify a safe reference set $R$ whose output $\mathcal{F}(R)$ is highly likely to satisfy the desired trust properties. For example, if $\mathcal{F}$ is a standard deep learning algorithm and $R$ is a clean dataset, then the trained model $\mathcal{F}(R)$ is likely robust to backdoor attacks.
>
> - We then control distinguishability by adding minimal noise to reduce the statistical divergence between the target model and the reference model. This provably ensures that the target model has a high probability of satisfying the desired trustworthiness properties (see Lemma 2.3 and Section 3.2(i)). More importantly, unlike DP, which adds noise isotropically, DSI selectively adds noises only in necessary directions to minimize utility overhead.
>
> DSI also enjoys operational benefits, which enables black-box algorithm analysis and the noise only needs to be added to the final output of a data processing procedure.
>
> **2. Additional Experiments on Copyright/Contribution**
>
> We have added a set of experiments for the copyright or contribution control using our DSI framework. Please refer to **Experiment 1 in attachment (https://anonymous.4open.science/r/4722-1FD5).** In this experiment, we finetune a stable diffusion model (v1-4 from https://huggingface.co/docs/diffusers/v0.8.0/en/training/text2image) on a collection $U$ of 425 paintings from 10 artists. We select the reference sets $R_i$ as $U$ excluding one painting, i.e., leaving-one subsets. Thus, the $(\epsilon, \delta)$ DSI parameters characterize the **per-sample contribution** to the trained model.
>
> In the figure, we present: (1) the original painting, (2) the generated image from the pre-trained diffusion model **before** fine-tuning, (3) the generated image **after fine-tuning without noise**, and (4) the generated images **after fine-tuning with DSI noise** in different indistinguishability budgets ($\epsilon$). From the plot, it is easy to observe that with a larger $\epsilon$, the fine-tuned diffusion learns the art style better and the generated pictures look more similar to the training data.
>
> In practice, the DSI parameters can be used to quantify and determine the contribution of each sample or data source, which provides economic solutions to data usage in collaborative learning.
>
> Finally, we highly appreciate if you could let us know whether we addressed your concerns or you have additional questions.
>
> [1] Zhai, Shengfang, et al. "NCL: Textual Backdoor Defense Using Noise-Augmented Contrastive Learning."
>
> [2] Cohen, Jeremy, Elan Rosenfeld, and Zico Kolter. "Certified Adversarial Robustness via Randomized Smoothing."
>
> [3] Aerni, Michael, Jie Zhang, and Florian Tramèr. "Evaluations of Machine Learning Privacy Defenses Are Misleading."

---

### Official Review · Reviewer_53pn · 2025-03-21

**Overall Recommendation:** 3

**Summary:**

The idea of DSI is equivalent to per-instance privacy (see "Y.-X. Wang. Per-instance differential privacy. Journal of Privacy and
Confidentiality," or the equivalent individual privacy of "V. Feldman and T. Zrnic. Individual privacy accounting via a renyi filter. Advances  in Neural Information Processing Systems") when the reference is between two datasets and removing a datapoint. The case of two arbitrary datasets was also explored in "Thudi, Anvith, et al. "Gradients look alike: Sensitivity is often overestimated in {DP-SGD}". No mention of
these existing and related works is mentioned in the paper, and differentiating to them is recommended.

On the technical aspects, unlike the past work the paper notes when just focusing on the Gaussian mechanism (and not the sampled Gaussian mechanism which could complicate the analysis and was analyzed in the aforementioned prior work) one can optimize the returned noise vector given the subspace of differences. This seems like the main novel
contribution.On the composition theorem, the method may be improved with the expectation based composition theorem of "Gradients look alike: Sensitivity is often overestimated in {DP-SGD}" which would seem to imply it's enough for each step to satisfy a $\gamma$ constraint.

**Claims And Evidence:**

Yes

**Essential References Not Discussed:**

See above

**Experimental Designs Or Analyses:**

Yes, it maybe better to use more recent baselines, e.g. from https://arxiv.org/pdf/2204.13650

**Methods And Evaluation Criteria:**

Yes

**Other Comments Or Suggestions:**

-

**Other Strengths And Weaknesses:**

-

**Questions For Authors:**

Please compare to the papers mentioned above.

**Relation To Broader Scientific Literature:**

Some related work is not mentioned, see above

After rebuttal:

I think prior work captured some of what is discussed in the paper. Conceptually, it was already discussed that input dependence is ok for per-instance privacy for certain applications (e.g. see unlearning and memorization implications from Thudi et al.). Operationally, I understand the novelty and the setting -- optimizing noise for the specific datasets. I feel connection to prior work and equivalences need to be further explored.

**Theoretical Claims:**

Yes

---

> ### Author Rebuttal · Authors · 2025-04-01
>
> Thanks for your valuable comments and suggestions!
>
> **1. Comparison with (Per-Sample/Individual) Differential Privacy (DP)**
>
> We believe there are at least three fundamental differences between Differential Trustworthiness (DT) and Differential Privacy (DP).
>
> 1-1) First, on **motivation**, though both DT and DP leverage distinguishability or statistical divergence, they target very different problems. Distinguishability measures the closeness of two distributions, which can be interpreted in two ways:
> a) It is hard to distinguish the source of a randomly drawn sample; b) The behavior of two random variables, $a \sim A$ and $b \sim B$, is similar in probability: $\Pr(a \in O) \approx \Pr(b \in O)$ for any set $O$.
>
> Privacy (confidential protection) typically adopts a) to model preventing adversaries from recovering secrets from leakage. If the leakages from different secrets are indistinguishable, no useful information is revealed to an adversary.
>
> In contrast, DT adopts b), using divergence between a target and a set of safe references to characterize the probability that an algorithm’s output satisfies certain trust properties. Imagine, for a set of reference training datasets $R_1, R_2, \dots, R_m$, if each trained model $\mathcal{F}(R_i)$ has a 99\% probability of not memorizing sensitive information $q_i$, then the divergence between $\mathcal{F}(U)$ and each $\mathcal{F}(R_i)$ provides a bound on the probability that $\mathcal{F}(U)$ does not memorize all $q_i$ **simultaneously**.
>
> 1-2) **Conceptually**, DT justifies the role of data dependence in trust-preserving mechanisms, whereas this **remains a challenge in privacy**. If a specific sensitive data point $x_0$ becomes a parameter in a privacy guarantee (e.g., achieving $(\epsilon, \delta)$-DP for the membership of $x_0$), this statement itself already leaks information about $x_0$. Section 1.2 discusses why input independence is essential for privacy and why most DP mechanisms rely on worst-case sensitivity.
>
> Since DP mechanisms calibrate noise based on the worst case, regular (average-case) data points may enjoy stronger privacy guarantees—motivating prior work on per-sample [1,2] or individual privacy [3]. However, formally quantifying this average-case amplification from the provable, global security parameters remains an **open** question. Existing works [1,2,3] can only estimate per-sample/individual privacy loss, which itself is sensitive and cannot be disclosed.
>
> A key insight from DT is that many trust concerns focus on the use (e.g. copyright) or influence (e.g. backdoor) of public data to train a model, or governing models (e.g. memorization) to prevent certain behaviors, and it suffices to ensure indistinguishability with respect to safe reference models. More importantly, unlike privacy-preserving operations, when leakage is not a primary concern, input-independence is **not** necessary, and thus we propose the more efficient DSI.
>
> Another distinction is that DT only requires one-way divergence—specifically, the $f$-divergence from the target output $\mathcal{F}(U)$ to the reference outputs $\mathcal{F}(R_i)$. In contrast, DP treats two adjacent datasets $X$ and $X'$ symmetrically, requiring two-way divergence to prevent leakage. See Section 3.2(i) for further details.
>
> 1-3). **Operationally**, we demonstrate how to determine the optimal anisotropic noise to achieve the required DSI guarantees. However, isotropic noise in prior works [1,2,3] cannot provide **controllable** per-sample/individual privacy loss. Moreover, our composition theorem (Theorem 3.4) is tight, whereas the composition approach in [2] involves complex expectation-based estimates, and [3] still relies on worst-case individual guarantees.
>
> We will incorporate those comparisons in our revision.
>
> **2. Comparison with State-of-the-art Results**
>
> Please refer to Table 1. We **have** compared with the DP-SGD benchmarks (De et al., 2022, https://arxiv.org/pdf/2204.13650) you suggested. By optimizing DSI noise, we achieve **7-10\%** improvement in test accuracy in all cases with varying indistinguishability budget.
>
> **3. Subsampling Amplification**
>
> We appreciate your insightful comments and acknowledge that our current results do not exploit subsampling-based randomness, which could further enhance the trust-utility tradeoff—particularly in applications like SGD. We highlight this as a promising future direction in Section 6. But it is also noteworthy that even **without** subsampling amplification, DSI mechanisms already significantly outperform best-known DP methods that has incorporated subsampling (Table 1).
>
> Finally, we highly appreciate if you could let us know whether we addressed your concerns or you have additional questions.
>
> [1] Y.-X. Wang. Per-instance differential privacy.
>
> [2] V. Feldman and T. Zrnic, Individual privacy accounting via a Renyi filter.
>
> [3] Thudi, Anvith, et al. "Gradients look alike: Sensitivity is often overestimated in DP-SGD.

---

### Decision · Program_Chairs · 2025-05-01

**Decision:**

Accept (poster)

**Comment:**

This work introduces Data‑Specific Indistinguishability, a framework for “differential trustworthiness” that relaxes the worst‑case guarantees of classical Differential Privacy (DP) into data‑dependent indistinguishability bounds. There are a number of key contributions of this work: (1) it formalizes DSI, derives an optimal anisotropic Gaussian mechanism, and proves composition and grouping theorems tailored to one‑way divergence against a set of “safe” reference models. (2) it shows how to integrate DSI noise into black‑box deep learning training without clipping bias, optimizing noise directions to tightly control trust‑related risks.(3) the paper demonstrates DSI’s efficiency/utility trade‑off on (a) memorization mitigation in LLM fine‑tuning, (b) backdoor defense in federated learning, and (c) copyright/contribution control in diffusion models, outperforming standard DP‑SGD at matching indistinguishability budgets. Overall the reviewers find the contributions interesting and solid, therefore I decide to accept the paper, but I would also encourage the authors to fully address the reviewers' additional requests.